# Eustatic, Climatic and Tectonic Controls on the Evolution of a Middle to Late Holocene Coastal Dune System in Shimokita, Northeast Japan

**Koji Minoura [1] and Norihiro Nakamura [2],***

[1]  Department of Earth Science, Tohoku University, Sendai 980-8578, Japan; sswxp736@ybb.ne.jp
[2]  Institute for Excellence in Higher Education, Tohoku University, Sendai 980-8576, Japan
*  Correspondence: n-naka@tohoku.ac.jp; Tel.: +81-22-795-3397

**Abstract:** The Pacific coast of the Shimokita Peninsula, Northeast Japan, is occupied by one of the larger dune complexes in Japan. This coastal aeolian dune complex developed during the Holocene in a monsoon-influenced temperate climatic belt. The stratigraphic and sedimentological characteristics of outcrops, exposures and cores indicate that four generation of aeolian dunes are presented. These dunes developed during eustatic regression following the post-glacial sea-level highstand. Seaward shoreline movement, combined with strong winds from the Pacific Ocean, enhanced aeolian grain transport on the beach, resulting in the onset of dune growth and the consequent shrinkage of the coastal forest. Northeast Japan is located in a transitional zone affected largely by monsoonal circulation from Siberia and Southeast Asia. Thus, the regional climate is responsible for atmospheric changes on a hemispheric scale. Intensified monsoons contributed to flooding produced by rains and snow-melt. Steep increases in annual precipitation at 7200–6300, 4700–3600, 3050–2500, 1850–1100, and 550–200 calendar years before present (cal. yr. BP) increased the amount of surface erosion, causing a large volume of sediment discharge toward the coast. Shimokita has experienced frequent earthquakes and tsunamis, which have reduced dune landform relief by sediment displacement.

**Keywords:** aeolian dune; climate fluctuation; eustatic change; Japan; Holocene

## 1. Introduction

The growth of coastal dunes has generally been ascribed to marine transgression-regression, wind strength, and sand supply [1–5]. During sea-level rise, inshore currents and waves may transport sand-sized grains in lowstand deposits ashore where the grains become available for aeolian reworking, whereas during sea-level fall marine and marginal-margin sand may become subaerially exposed and available for aeolian transport [3,6–8]. Sand supply is essential for a beach to form [2,9], and beaches may be exposed to aeolian erosion by prevailing onshore winds [10–13]. Beach exposure and winds that are strong enough to entrain sand grains may result in a landward transport of wind-driven grains from the littoral zone off the low water line at the shoreface [6,7,13,14]. Thus, an abundant sand supply from the surf zone may increase the potential of dune development [2,4,15]. The role of water table responses to sea level changes is also an important factor that influences coastal dune systems [16].

Modern coastal zones are generally regarded as fragile environments [17–20], and atmospheric conditions may exert considerable influences on the development of coastal dunes [21–23]. Weathering may solidify or stabilize a dune surface through microbiological and biochemical activity [4,24], and coastal dunes stratigraphy may reveal evidence of previous episodes of soil formation and dune stabilization [10,25]. The shapes of dune bodies are regulated by the patterns of airflow, sand supply, and precipitation [26–28], which are under the control of the climate [23,24,29].

As the sea-level reached a highstand during the middle Holocene, longshore currents acted as an important agent of sediment reworking and redistribution on the shoreface at many locations around the world [30–32]. Onshore sand migration, wind strength/direction, and ground surface condition controlled the growth process and form of middle to late Holocene coastal dunes [6,33–35]. In Japan, coastal dune complexes are well developed along the Pacific coast of the Shimokita Peninsula, Northeast (NE) Japan. The Shimokita Peninsula is within a climatic zone affected by two different monsoonal circulations originating in Southeast (SE) Asia and Siberia [36]. Thus, warm/wet or cool/dry air masses likely exert a far-reaching influence on the regional surface environments in response to seasonal fluctuations [37]. The Pacific coast of the Shimokita peninsula is the easternmost emerged part of the arc-forearc system facing the Japan trench [38] (Figure 1A), and the coastal belt is exposed directly to tectonic and oceanic events [39].

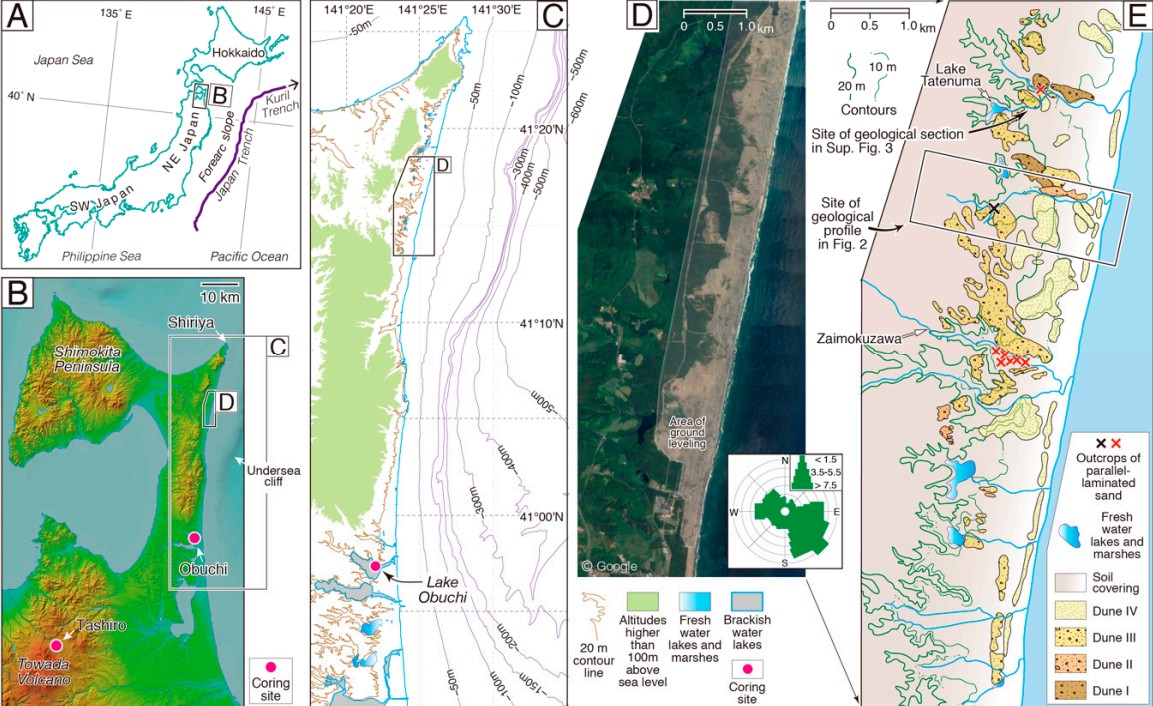

**Figure 1.** (**A**) Tectonic division of NE Japan, including, from east to west, the Japan Trench, NE Japan Arc, and Japan Sea. The study area is shown by the inserted square. (**B**) Relief map showing the entire region of the Shimokita Peninsula. The map issued by the Geographical Survey Institute of Japan is utilized for displaying the research area and the coring sites (marked in red). (**C**) Topographic map of the coastal zone and the undersea slope. The mid-late Holocene dunes extend along the Pacific coast of the Shimokita Peninsula for more than 45 km. (**D**) Google satellite image of the Shimokita Peninsula. (**E**) Geological map of the coastal dune field of the Shimokita Peninsula, following [40]. The red crosses denote the major outcrops of parallel-laminated/bedded sand. The black cross indicates the occurrence of the tsunami deposit of the 12th century. The open arrows show the sample locality of standing fossil conifer for accelerator mass spectrometry (AMS) age determination (Tatenuma and Zaimokuzawa).

This study presents a detailed reconstruction of lacustrine processes with the objective of clarifying the atmospheric and oceanic functions on the evolution of coastal dunes of the Shimokita Peninsula. In Japan, it is generally accepted that the coastal dune buildup occurred along with marine regression when the sea-level fell [41–43]. The coastal dunes of the Peninsula are generally thought to have formed after the Middle Holocene Climatic Optimum [44], and their origin and behavior have been controlled by shoreline sediment and current movement and by atmospheric changes [45–47]. In addition, the dunes have been activated as a result of winter monsoon intensification since the beginning of the Middle Holocene [34,48]. Greater detail of these processes and changes are described in this paper

using new data from a brackish lake associated with the coastal dune field and from a wetland in a highland area approximately 60 km southwest of the dune field study site.

## 2. Study Area

The primary study area is the Pacific coast of the Shimokita Peninsula (Aomori Prefecture, northeast Japan) where there is a Holocene dune complex with a large variety of surface features, such as lakes, rivers, marshes, and vegetation [49]. This dune complex is of Holocene age, and is one of the larger fields of aeolian dunes in Japan. For this study, the field area is the northern coast (Figure 1B,C), where the dune surface is covered with extensive pine trees, that were cultivated during the early 20th century for protection against wind erosion [45]. Under the present surface conditions, grain transport by winds through saltation or creeping is observed on the exposed beach and backshore dunes, mainly during the winter season. In former times, the dunes (sandhills) were extremely deficient in vegetation cover, and the dunes were exposed to intense erosion by monsoonal storms or torrential rains [50,51].

The presence of thick vegetation prevented a clear view of the topographic situation of the sandhills. The repeated development of soils has led to the loss of stratigraphic information on aeolian development [44]. Together with the stratigraphic complexity, the difficult field conditions limited the geological study on the origin of dunes, and the identification of dune types remains uncertain. A macro-scale structure showing a negative slope toward the slip face is typical of U-shaped sand dunes [52]. A concave-downward shape of beds and laminae is characteristic of the sand sequences exposed at Tatenuma and Zaimokuzawa (Figure 1E), and thus, the observed sand dunes are considered in this paper to be parabolic dunes [53]. The low-angled bedding was likely formed on the slip face of the downwind-side sand bedforms (dunes). Tanino et al. [34] reported the genesis of parabolic dunes at the northern end of the Peninsula. The general trend of dunes facing NW–SE is parallel to the prevailing east-southeast to west-northwest winds in this area (wind roses in Figure 1D; The Meteorological Agency of Japan, www.jma.go.jp/). Such winds are typical during the winter monsoon in Shimokita (e.g., [54]).

A secondary study area is Lake Obuchi located at the southern end of Shimokita Peninsula, which is 30 km south from the dune field study area. Lake Obuchi, which has an average depth of nearly 2.75 m, is a brackish-water, bell-shaped lake of 3 km long and 1.2 km wide with a small inflowing river from the west. Data from Lake Obuchi were used to obtain time-continuous and high-resolution records of the chemical, sedimentological, and paleontological properties.

A tertiary study area is the Tashiro Wetland located in a highland area approximately 80 km southwest of the dune field study site (Figure 1B). The Tashiro wetland is located inside a volcanic crater in the Hakkoda volcano that formed ~16,000 years ago [55]. The Tashiro Wetland has no inflowing or outflowing rivers, the basin where the Tashiro Wetland is located has been isolated from the surrounding environments by caldera walls [56], and the vegetation inside the basin has been the main supplier of pollen grains to the wetland. Data from the Tashiro Wetland were used to obtain a time-continuous and high-resolution record of atmospheric variability on the basis of pollen stratigraphy, to be used for comparison and interpretation of data from the coast of the Shimokita Peninsula.

The westerly winds blow from the west to the east across the Asian Continent (e.g., [57]). This meandering wind is a major factor in the longitudinal shifting of atmospheric circulation in the middle latitudes [58]. During the high winter months (December–February), the northeastern part of the Eurasian Continent experiences enormous barometric pressures (Siberian High), which frequently extends to the northwest Pacific [54]. Under the expansion of continental air masses, strong cold fronts easily reach the Japanese Islands [57]. The high-pressure system propagates cold and dry air [54], which increases in moisture content during the course of migration above the Japan Sea [59]. The humid air masses produce dense snow clouds on and above the Japanese Islands (Japan Meteorological Agency; www.jma.go.jp/). As a result, cold districts experience tremendous snowfalls [60]. The Tsushima Warm Current flows northward across the Japan Sea and largely contributes moisture to the air mass [61].

The volume of rainwater under the modern weather conditions on the Shimokita Peninsula, at an annual mean of ~440 mm, is very low (www.jma.go.jp/), and the drainage systems are poorly developed along the coast. If the growth of snow clouds is activated under expanded monsoonal circulation, then the coastal zone might be covered in much thicker snowfall than in the existing situation. Rapid seasonal thawing triggers floods of melted snow, which cause large drainage channels to appear in the dune field. A considerable part of the terrestrial precipitation in Japan is released to the sea as seabed groundwater discharge [62].

The NE Japan arc exists at the northwestern Pacific convergent plate boundary, where the oceanic plate descends under the edge of the arc crust. Under these tectonic conditions, the Shimokita Peninsula is subjected to E–W compressive stress. Therefore, the Shimokita Peninsula faces the north–south extension of the narrow shelf to the east. Precipitous marine cliffs, arranged parallel to the coastline, border the shelf margin (Figure 1B,C). The forearc slope extends from the cliff bottom as far as the Japan Trench (see arrow indicated "undersea cliff" in Figure 1B), and the wide and gentle slope is stepped by slip plains running parallel to the trench axis [63]. During the middle to the late Holocene, the Pacific coast of the Shimokita Peninsula has been scarcely subjected to tectonic elevation or subsidence (The Japan Atomic Power Company; https://www.nsr.go.jp/data/000153120.pdf). Tectonic movement during Holocene has been negligible in the Shimokita Peninsula [64].

The Pacific coast of the Shimokita Peninsula has a history of repeated tsunami disasters [65], and it can be estimated that the sediments from the lake preserve detailed records of tsunami events. The 2011 Great Earthquake tsunami reached the Pacific coast of the Shimokita Peninsula about 40 min (min) after the main shock, with waves 4–5 m in height [66]. The run-up currents passed through a narrow waterway and flowed into Lake Obuchi. Fishermen provided eyewitness accounts of the lake water becoming cloudy as the seawater rushed in.

## 3. Methods

Field work was conducted along deep streams, focusing on the stratigraphic and topographic development of aeolian dune deposits and soil layers (paleosols). Approximately 250 arenaceous sediment samples were collected from the un-weathered horizons of sand dunes. After removing organic carbon and carbonate, the residues of each sample were measured using an automatic grain-size analyzer with a laser diffraction system (SALD 3000J; Shimadzu Corporation, Kyoto City, Japan). The modal compositions of sand were analyzed only for the medium-grained fraction, in order to reduce the effect of grain-size variation (e.g., [67]). Modal counts were done of about 300 grains using a polarization-microscope. Diatom valves were identified in some aeolian dune sediments and paleosols, following the method described by [68].

To identify the terrestrial and marine influences on the evolution of the dunes, drilling was performed in Lake Obuchi (Figure 1B,C; 40°57.47′ N, 141°21.73′ E). Drilling was performed to a sediment depth of 10 m in November 2012 by employing thin-walled steel pipes. Prior to the core recovery, the bottom surface of the lake was dredged by using an Ekman-Berge bottom sampler. The living shells of *A. beccarii* were also collected by employing Ekman-Berge bottom samplers.

The Lake Obuchi core was sliced at 1 cm intervals, and each separated specimen was submitted for conventional smear slide analysis (Figure 2). Owing to the lack of meaningful geological information in the lower part of the core, which was composed of homogenous sand with no mud seams or fossils, this study focused on the upper half (450–0 cm drilling depths) for sedimentological (sediment fabric and grain composition), mineralogical (mineral properties and composition), paleontological (benthic and planktic foraminiferal assemblages), and limnological (oxygen and carbon isotope chemistry) data after measuring wet and dry densities. After removing the organic carbon and carbonates from each sliced sample by hydrogen oxide and hydrochloric oxide treatments, respectively, the grain size composition of the residues was measured using an automatic grain-size analyzer with a laser diffraction system (SALD 3100; Shimadzu Corporation, Kyoto City, Japan). The mud content and median diameter were calibrated from the output data of grain size measurement. The mineralogy

of grains coarser than 63 μm was investigated under an optical microscope with a polarizing plate. To recover foraminiferal fossils from the Lake Obuchi core, sediment samples were dried and then sliced at room temperature, and the fraction of grains coarser than 63 μm was isolated through a 230-mesh sieve. All the foraminiferal shells from each of the treated samples were isolated and examined under a stereoscopic binocular microscope.

Oxygen and carbon isotope analyses were performed on monospecific foraminiferal shells (*Ammonia beccarii*). Fossil and living foraminifera were analyzed using a mass spectrometer (Delta V advantage; Thermo Fisher Scientific). A sub-sample of about 0.10 mg was reacted with $H_3PO_4$ with a specific gravity of 1.92 in a vacuum at a constant temperature of 70 °C. Isotope ratios were corrected for $^{17}O$ interference with the equations presented by Santrock et al. [69]. Results are reported in the conventional δ–notation relative to the Vienna Peedee Belemnite (VPDB) international standard, which was cross-checked against the NBS-19 international standard. The precision, as deduced from daily replicate measurements of an internal laboratory calcite standard (MACS1; $δ^{13}C$ = 1.18‰ VPDB, $δ^{18}O$ = −4.72‰ VPDB), was better than 0.06‰ for $δ^{13}C$ and 0.07‰ for $δ^{18}O$.

At Tashiro Wetland, a core 8.80 m in length was drilled in the center of the crater at 41°57.04′ N and 141°29.62′ E, of the altitude 570 m, using a thin-walled core sampler (Figure 3). The core was cut into 1 cm intervals, and 790 samples were investigated for sedimentological (grain composition and size) and biological (pollen and plant opal) data after measuring the wet and dry densities. For fossil pollen extraction, a standard pre-treatment procedure was used, including KOH–HF treatment and heavy liquid separation by acetolysis [70]. Approximately 250 fossil pollen grains or more were examined for each sample slide under a high-powered microscope.

For identifying the fallout tephra found at a depth of 287–288 cm of the Lake Obuchi core (Figure 2A), the chemistry of glass shards from the tephra flakes was analyzed using a wave-dispersive electron probe micro-analyzer (WD-EPMA), JEOL Ltd., Tokyo, Japan.

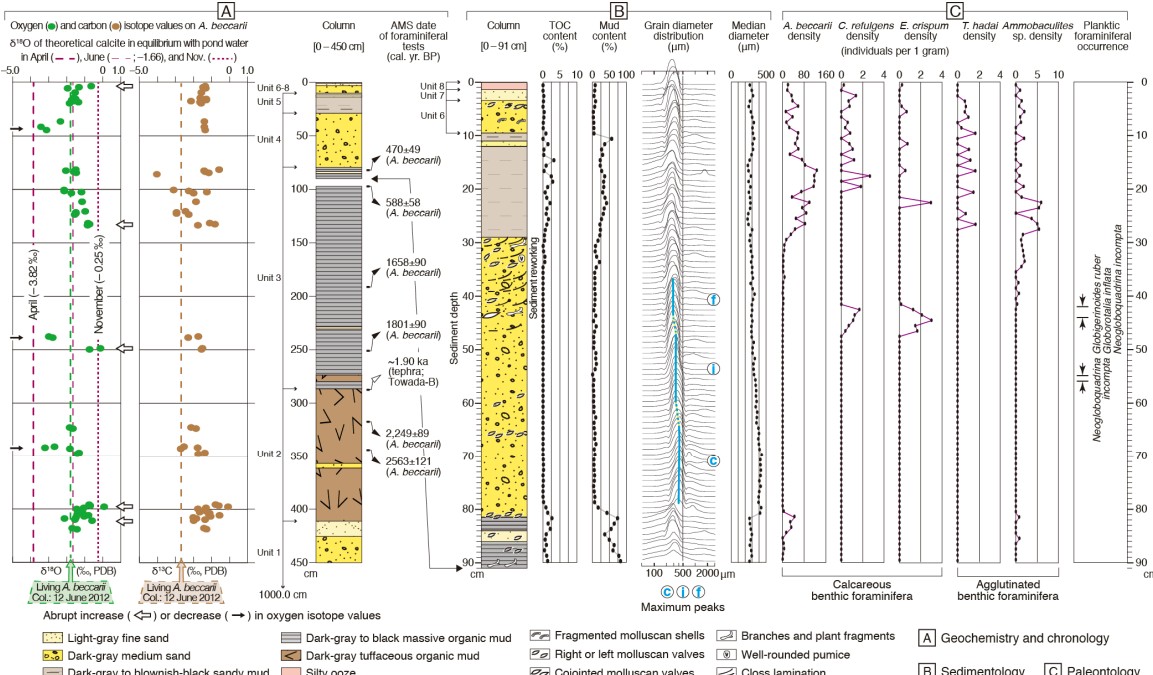

**Figure 2.** Correlation of the stratigraphic changes in the chemical, sedimentological, and paleontological properties of the Obuchi core. The depth profiles show, from left to right, the fluctuations in (**A**) foraminiferal stable isotope values; (**B**) organic matter content, mud content, and grain size distribution; (**C**) foraminiferal occurrence. The isotopic results regarding fossil foraminifera, living foraminifera, and pond water are listed in Supplementary Table S3-1, S3-2, and S3-3, respectively. The seasonal fluctuations in calcite oxygen isotopes were theoretically estimated by using data of water

temperature and salinity [71]. The depth profiles of mud and organic matter contents at the depth interval between 90 and 0 cm show a positive correlation between them. The succession of maximum peaks that appear in the stack of output profiles of grain size is divided into lower coarse (**c**), middle intermediate (**i**), and upper fine (**f**) columns. The chronological results in column A are listed in Supplementary Table S4-1.

By applying the best modern analog (BMA) method for the fossil pollen data, it is possible to reconstruct Holocene paleoclimate conditions in the Shimokita Peninsula. Using this numerical technique, it is assumed that pollen assemblages with a similar composition of taxa were produced by similar flora, regarding structural and compositional aspects, and that each flora grew under a similar climatic condition [56]. This assumption enables the identification of the closest modern analogs for each analyzed fossil sample by comparison with modern pollen samples selected from an appropriate reference dataset. The modern climate parameters around the sites of modern pollen samples may then be assigned to the analyzed fossil samples and used to reconstruct values of the past climate [72].

With the objective of reconstructing the past climate of the Shimokita Peninsula, the free-access software package Polygon 1.5 [73] was applied to a dataset of surface pollen spectra from 285 sites in Japan [74] and a dataset of modern climate at the surface pollen sites [75]. In total, 327 datasets were selected from the compiled palynological results for the calculation (Supplementary Table S5). The pollen density of the slides from the unselected samples was nearly two-thirds of that of the selected samples. The wider-ranging microscopic observation arising from diluted grain distribution in a sample slide likely produced some difference in the statistical quality of pollen-grain measurement against the selected samples. To determine the similarity between each pollen spectrum in this calculation, chord distance, which is a Euclidian metric between two points in the *n*-dimension space defined by the square root of the pollen percentages, was adopted. Eight spectra having the smallest chord distance were selected as the BMAs of the analyzed pollen spectrum. The computational output on annual mean air temperature and annual precipitation are listed in Supplementary Table S6. The accuracy of the reconstructed climate parameters can be evaluated by the correlation coefficients between the calculated value and the estimated modern climate at each surface pollen site by using the leave-one-out method. Accuracies were calculated to be 0.89 for annual mean temperature and 0.70 for annual precipitation, which are considered to be fairly high.

Accelerator mass spectrometry (AMS) age data of the Tashiro core reported by Shinozaki et al. [37] were used to estimate the chronological transition of the climate parameters (Figure 3). The radiocarbon ages were reported in radiocarbon ($^{14}$C) years before present (BP) with a 2-sigma (2σ) standard deviation of the age uncertainty, using the Libby half-life of 5568 years (yr) with 0 $^{14}$C yr BP being equivalent to AD1950. The radiocarbon ($^{14}$C) ages of 42 samples were then calibrated with the CALIB 5 program [76] to calendar years (cal. yr.) BP using the INTCAL13 calibration dataset [77]. Figure 3B shows age (in cal. yr. BP) versus depth plots, which include the chronological data of the tephra beds. To ensure that the detailed chronological fluctuations in temperature and precipitation were included, sedimentation rates were calculated at every depth interval of the age data. The date of each statistical output was determined on the basis of the sedimentation rate of a corresponding horizon (Supplementary Table S6).

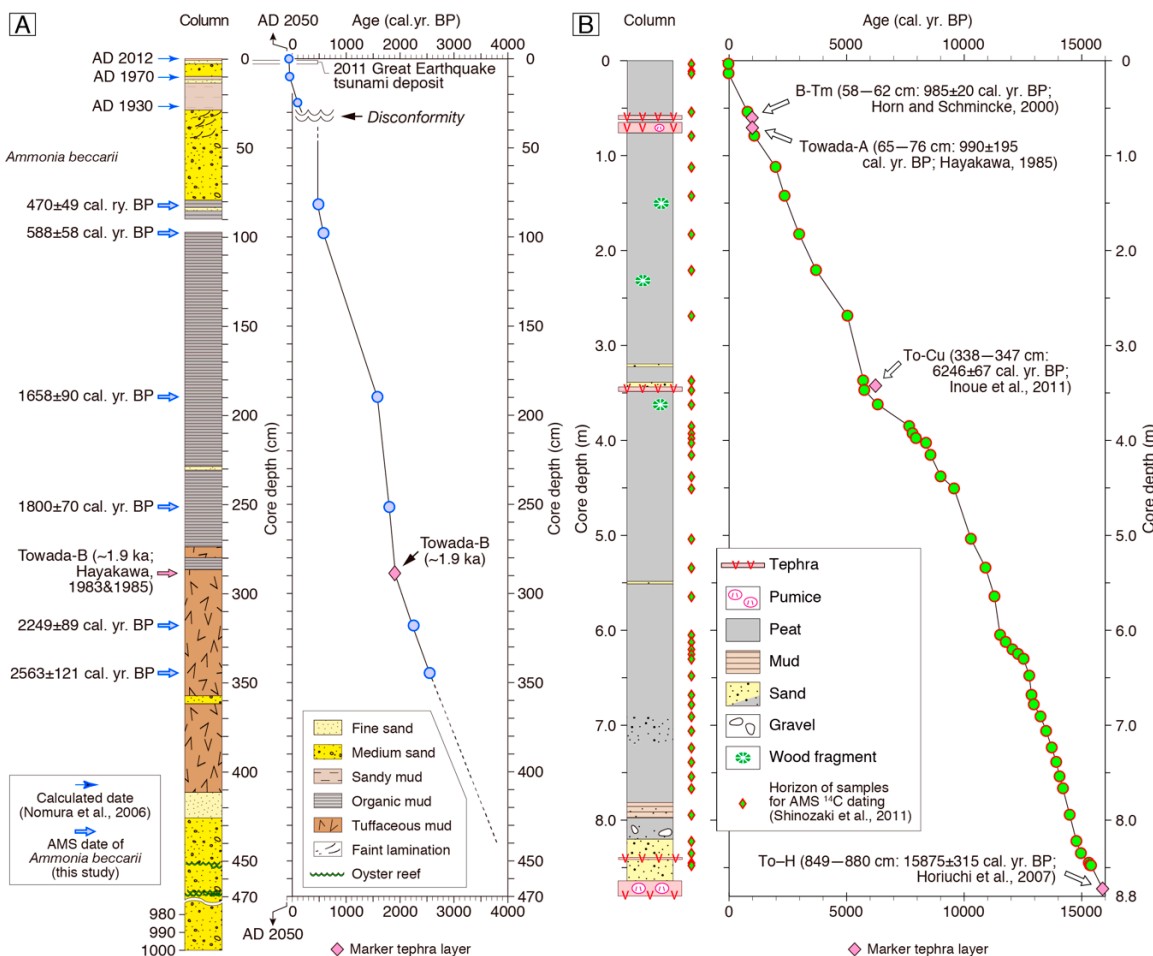

**Figure 3.** AMS age (cal. yr. BP) versus sediment depth (cm) plots relative to the cores from (**A**) Lake Obuchi and (**B**) Tashiro Wetland. Each column shows a lithological profile of the core. The chronological results in column A are listed in Supplementary Table S4-1. The AMS dates and the tephra chronology of the Tashiro core are from Shinozaki et al. [37].

## 4. Modern Depositional Environments of the Shimokita Peninsula

The study area of the Shimokita Peninsula contains the following modern geomorphological systems: (1) beach environment; (2) coastal aeolian dunes; (3) lacustrine environments; (4) fluvial environments; and (5) alluvial fans.

### 4.1. Beach Environment

The northern coast has a ~15 km long and 100–150-meter-wide sand beach with small (less than 2 m high) backshore dunes and freshwater marshes behind it. Shrubs and evergreens cover the ridges and slopes of the backshore dunes (Figure 1D), vegetation is present only sporadically along the valleys cutting through the area, and vegetation is absent from the beach environment. Coastal lakes receive water from land and sea, and thus, lacustrine conditions are influenced by both precipitation and marine condition (e.g., [78]). Quite often, seawater by seasonal coastal flooding reaches the back-marsh areas. The prevailing winds during the winter are directed onshore to WNW, and the resultant drift moves grains landward on the exposed shore. The modern riverbed dips gently to the east (Figure 4), and some water systems connect isolated lakes with the Pacific (Figure 1E). The modern beach of the Shimokita Peninsula consists of medium sand that is composed of feldspar (50–60%), quartz (25–30%), heavy minerals (10–20%), rock fragments (2–4%), and biogenic grains (1–3%). In total, 80–90% of heavy

minerals are enstatite and augite. The biogenic grains consists of freshwater diatom fragments and marine diatom fragments (<2%), and a few unidentifiable biogenic grains (3–4%).

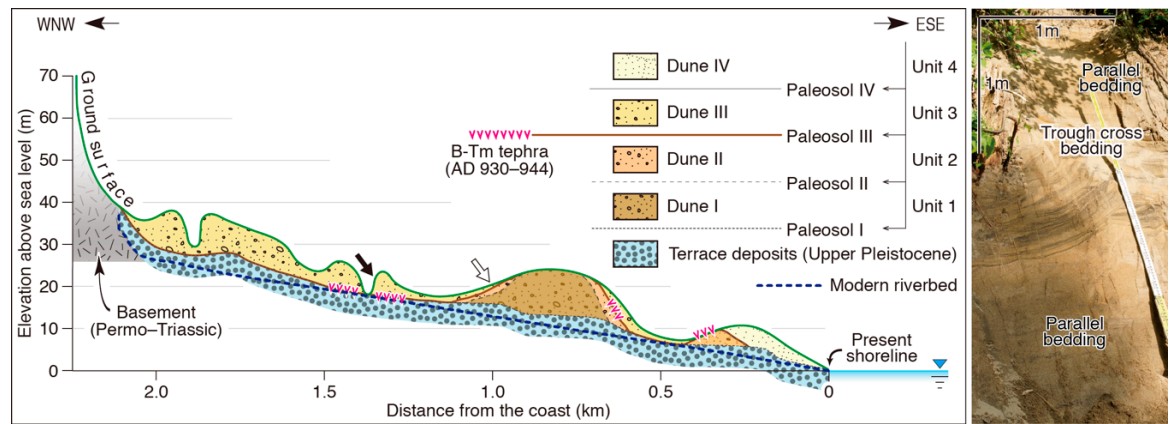

**Figure 4.** Geological cross-section along the box in Figure 1E. On the basis of the formation of soil layers (Paleosol I to IV), the dunes are classified into four stratigraphic units: Dune I, 6000–4000 cal. yr. BP; Dune II, 2000–1500 cal. yr. BP; Dune III, 1000–500 cal. yr. BP; and Dune IV, 150–60 cal. yr. BP. A panoramic view of an outcrop is displayed on the right, showing the depositional sequence of parallel bedded and cross-laminated feldspathic sand (a downward solid arrow in the left denotes the outcrop location). The soil layer of Paleosol III is well developed and traceable throughout the dune field. The open arrow shows the sampling site of the aeolian sand for diatom analysis.

*4.2. Coastal Aeolian Dunes*

The coastal aeolian dune field is ~30 km long and 1.5–2.5 km wide (Figure 1C,D). The inland (western) boundary of the dune field is an irregular narrow zone (50–200 m in width) defined by the eastern terminations of alluvial (talus) sediments. Miocene volcanic rocks (Tomari Formation; [79]) exposed in the upstream (western) part of the dune field [80]. Most of the dunes are vegetated, and only the nearshore (eastern) part of the dune field is not vegetated. The dunes form 0.5–1.5 km long linear ridges that trend NW to SE (Figure 1E). Relief in the dune field ranges from 5 to 15 m, and the crest of the dunes reach a maximum height of 39 m above sea level. The aeolian dunes rest on an unconformity above the Upper Pleistocene of gravels and sands that form terrace deposits (Figure 4).

The run-up currents of the 2011 great earthquake tsunami inundated the coast and deeply cut into the exposed dune bodies. One exposure in this area was described in detail at 41°17′19.8744″ N, 141°24′14.5254″ E. This exposure was 27.5 m high, and was divided into four stratigraphic units on the basis of lithological composition (Unit 1 to Unit 4; Figure 4). Each unit consists of cross-laminated medium sand overlain by organic soil without obvious sedimentary structures. The lowest unit (Unit 1) lies above an unconformity that caps poorly bedded cobble-pebble gravels and sheet-like bands of sand with cross-bedding. These stratigraphic units are described (from base to top) as follows.

Unit 1: Unit 1 consists of 10–14 m thick pale-orange colored medium sands (Dune I) and underlying 2.0–3.0 m thick grayish-brown clayey organic silts (Paleosol I). The sands are composed of 20–25% quartz, 55–60% feldspar, 10–15% heavy minerals, ~1% rock fragments, and 1–2% biogenic grains. Pristine frustules of *Achnanthes bahusiensis*, *Achnanthes minutissima*, *Cyclotella striata*, *Eunotia implicata*, *Gomphonema parvulum*, and *Hantzschia distinctepunctata* are present in the sands (Supplementary Figure S1). Sedimentary structures include planar and trough cross-bedding, and concave-downward fore sets in the cross-bedded sands.

Unit 2: Unit 2, which overlies Unit 1, consists of ~10 m thick pale-orange medium sands (Dune II) and underlying 1.5–2 m thick gray-brown clayey silts (Paleosol II). The sands are composed of 20–30% quartz, ~60% feldspar, 10–15% heavy minerals, ~1% rock fragments, and 2–3% biogenic grains. The silts contain standing fossil trunks and roots of *Thujopsis dolabrata* var. Hondai. The sands

include pristine frustules of *Achnanthes bahusiensis*, *Achnanthes lanceolata*, *Cyclotella striata*, *Eunotia bilunaris*, *Eunotia pectinalis*, and *Gomphonema parvulum* (Supplementary Figure S1). Sedimentary structures are mostly planar and trough cross-bedding, and concave-downward laminae are rather rare.

Unit 3: Unit 3, which overlies Unit 2, consists of 10–15 m thick gray-yellow medium sands (Dune III) and underlying ~1.5 m thick gray clayey silts (Paleosol III). The sands are composed of 20–25% quartz, 50–60% feldspar, 10–15% heavy minerals, ~1% rock fragments, and ~1% biogenic grains. The sands include pristine frustules of *Achnanthes minutissima*, *Cyclotella comta*, *Cyclotella striata*, *Diploneis ovalis*, and *Gomphonema parvulum* (Supplementary Figure S1). Sedimentary structures are mostly planar and trough cross-bedding.

Unit 4: Unit 4, which overlies Unit 3, consists of ~10 m thick gray-white medium sands (Dune IV) and underlying ~1 m thick grayish clayey silt (Paleosol IV). The sands are composed of 20–25% quartz, 55–60% feldspar, 10–15% heavy minerals, ~1% rock fragments, and 2–3% biogenic grains. The silts contain standing fossil trunks and roots of *Thujopsis dolabrata* var. Hondai. Pristine frustules of *Achnanthes bahusiensis*, *Achnanthes minutissima*, *Diploneis interrupta*, *Gomphonema parvulum*, *Hantzschia distinctepunctata*, and *Navicula clementis* are present within the sand. Sedimentary structures are mostly planar and trough cross-bedding, and concave-downward fore sets are present in cross-bedded sands.

These units are described as follows: (1) Dune IV and any immediately overlying modern vegetation, (2) Dune III and immediately overlying Paleosol IV, (3) Dune II and immediately overlying Paleosol III, (4) Dune I and immediately overlying Paleosol II, and (5) Terrace deposits and the immediately overlying Paleosol I. The dunes are formed generally of pale-yellow medium sands, which consist of feldspar and quartz grains with ferromagnesian minerals as accessory components. Concave-downward beds and laminae are visible in some vertical exposures at Tatenuma and Zaimokuzawa (see the crossed points in Figure 1E). Beds and laminae are defined by grain sorting and opaque-mineral accumulation. The maximum bed thickness is several meters and the maximum bed or laminae length is 8 m. The dip angles of most beds and laminae range between 15° and 20°. Only a few higher-angled beds are visible. The direction and spread of the dip measurements are NWN–SWS and 150–180°, respectively. Asymmetrical troughs are recognized in cross-bedded sand by the rhythmic interlayering of dark stripes arising from the concentration of ferromagnesian minerals (Figure 4). Pyroxenes are the major constituents among the dark stripes (Supplementary Figure S4), and the mineral composition is very similar to that of the beach sand.

According to radiocarbon ($^{14}$C) dating of annual tree rings and mineralogical analysis of tephra beds, Minoura et al. [40] classified the dunes into the following four stratigraphic units: Dune I, ~6000–4000 cal. yr. BP [maximum thickness: 12 m]; Dune II, 2000–1500 cal. yr. BP [maximum thickness: 4 m]; Dune III, 1000–500 cal. yr. BP [maximum thickness: 10 m]; and Dune IV, 150–60 cal. yr. BP [maximum thickness: 5 m]. Each of these dune units is capped by a disconformity defined by a well-developed humic soil layer. These humic soil layers are referred to as Paleosol I to IV, respectively. The soil layers form undulating slopes, and Paleosol III is traceable across the dune field (Figure 4). Fossil forests of standing trunks of *Thujopsis dolabrata* var. Hondai occur close together from the basal horizon of each dune unit (Dune I–Dune IV; Figure 1E). The trunks have sent down roots deep into the soils underlying the dunes. AMS ages of 292 ± 18 and 296 ± 16 cal. yr (with 2σ range of uncertainty) were obtained from the second and fourth outermost annual rings, respectively, of a fossil tree (*T. dolabrata* Hondai) collected from the flood plain deposit in Tatenuma (Figure 1D; Supplementary Figure S3A, Supplementary Table S4-2).

Twenty genera and 51 species of diatoms were identified in the aeolian sand samples of Dune III (Supplementary Figure S1). Freshwater species dominated the frustule composition of cross-bedded sand (~95%), and marine diatoms were rare (3–3.5%). The sand and gravel sediments beneath the dunes are composed of coarse to medium sand and pebble/cobble clasts in various proportions, with clasts more predominant in lower horizons. The sands show sub-horizontal beds often 30 to 40 cm thick.

Low-angle cross bedding is common in the beds. The sand and gravel sediments are unconformably overlain by the organic soils of Unit I (Paleosol I), which consists of massive clayey silt with plant leaves and roots. Restricted outcrops of gravelly sediment with trough-cross bedding (interpreted as fluvial deposits; see Section 4.4 below) are present throughout the dune field.

### 4.3. Lacustrine Environments

Several lakes are present in the study area along the western side of the aeolian dune field. The biggest lake is Lake Onuma, which is ~1.6 km long and ~300 m wide. Another lake is Lake Tatenuma, which is ~300 m long and ~100 m wide with a river that flows out to the east margin of the lake (site of geological section in Supplementary Figure S3).

The seasonal fluctuations in oxygen isotopes of Lake Obuchi were estimated by adopting the data on water temperature and salinity reported by [77]. The calculated results for $\delta^{18}O$ of all lake waters at 4 m water depth are listed in Supplementary Table S3-3, which includes the physical values (water temperature and salinity) used for the calculation. In all of the lakes, oxygen isotope ratio of living shells (−1.79‰ VPDB; Supplementary Table S3-2) was very close to that of the theoretical calcite (−1.66‰ VPDB; Supplementary Table S3-3) in isotopic equilibrium with the lacustrine water of Lake Obuchi in June (−1.64‰ Vienna Standard Mean Ocean Water (VSMOW); Supplementary Table S3-3). This closeness of values demonstrates that the $\delta^{18}O$ values of fossil shells are indicative of the aquatic conditions of habitats. The oxygen isotope values shown on the stratigraphic profiles in Figure 2A include considerable deviations from the trend shown by mean values. The prevalent biochemical conditions of Lake Obuchi fluctuate seasonally [81], and the increase in surface productivity during summer stagnates the lake bottom [77]. Under anaerobic conditions, the total organic carbon (TOC) content reaches up to 4% at the sediment/water interface [82].

### 4.4. Fluvial Systems

Several rivers flow across the northern-half area of the dune field, and extensive deposits of well-sorted feldspathic medium sand are present along these rivers. At Lake Tatenuma and River Zaimokuzawa, the valley floors are filled with sand, reaching up to 1.5 m in thickness (Figure 1E; red crosses). The sand, which contains abundant detrital grains of mafic minerals, overlies an undulating unconformity (with groove and scour structures) on top of Paleosol III (Supplementary Figure S2). This Paleosol III that is exposed along the fluvial channels (described in Section 4.4) is the same Paleosol III that is exposed among the coastal aeolian dunes (described in Section 4.2). The sediment above this unconformity is divided into a lower unit of sand without obvious sedimentary structures and an upper unit of laminated sand. The lower unit is 25–40 cm thick, and contains mud flakes and wood fragments. The upper unit is 95–110 cm thick, and contains water-escape structures such as flame-like pillars, dishes, and sand dikes, as well as traces of lamina truncation and sediment framework disturbances. The sedimentary structures of the upper unit also include rhythmic alternations of gray (felsic) and black (mafic) laminae. The dark (black) laminae are composed mainly of pyroxene and magnetite, whereas the light (gray) laminae are composed of feldspar and quartz. The diameters of the high-density grains are smaller than those of the less dense grains, and a pair of gray and black laminae resembles inverse grading. The felsic laminae contain abundant intact valves of *Achnanthes bahusiensis*, *Cyclotella striata*, and *Hantzschia distinctepunctata* (Supplementary Figure S1). Fossil trunks of *Thujopsis dolabrata* var. Hondai extend upward to the mud layer and are truncated by a bed of the sand without obvious sedimentary structures in the third unit (Supplementary Figure S2). At several locations, we found the intercalated deposition of feldspathic medium sand above the organic soils (Supplementary Figure S2). The sand is well-sorted and lacks silt and clay. Parallel bedding was observed on fresh exposures.

On the bank of the stream that flows east from Lake Tatenuma, there is an excellent exposure of sediments that display various styles of bedding and intraformational loose/soft-sediment deformation (Supplementary Figure S3A). The exposure is located 1.5 km inland from the present shoreline and at an

elevation of ~15 m (Figure 1E). The exposed succession consists of a lower unit with a 1.5–1.8 m thick bed of gravelly medium/coarse sand without obvious sedimentary structures. This lower unit is overlain by a 60–90 cm thick bed of medium sand with decimeter-scale trough cross-bedding (Supplementary Figure S3B). The unit of medium sand is overlain by a ~70 cm thick bed of humic silty mud (Paleosol III in Supplementary Figure S3A) containing fossils of deciduous tree leaves and herbaceous rootlets (Supplementary Figure S3C,D). Freshwater shrimp burrows (*Paratya improvisa* De Haan) are ubiquitous in the lower half of the lower unit of gravelly medium/coarse sand (Supplementary Figure S3A).

Several exposures of the distinctive sand, exhibiting a lower unit of sand without obvious sedimentary structures and an upper unit of laminated sand, are present along the rivers flowing across the northern area. Mud flakes (interpreted as rip-up clasts from the underlying soil layer, Paleosol III) occur within the basal part of the lower unit (gravelly medium/coarse sand without obvious sedimentary structures). Intra-formational deformation structures are present in the sand of Paleosol IV, which is the same as that is exposed among the coastal aeolian dunes (Supplementary Figure S3C,D).

Along the river with its source in Lake Tatenuma (Figure 1E), there is an exposure of the following units: (1) a 120 cm thick unit of gravelly sand without obvious sedimentary structures, overlying by (2) a 30 cm thick unit of sand with decimeter-scale trough cross-bedding, which is overlain by (3) a 40 cm thick unit of humic silty mud (Paleosol IV, Supplementary Figure S3A). Well-rounded pebbles and cobbles occur from the base of each trough cross-bedded sand in Paleosol IV (Supplementary Figure S3B). A unit of unstratified medium to coarse sand with well-rounded pebbles and cobbles is overlain by pebbly coarse sand (Supplementary Figure S3B). Gravel imbrication and scour and fill structures are common in the lower unit of gravelly sand but are missing in the upper unit of sand without obvious sedimentary structures. Underlying the sand without obvious sedimentary structures, a unit of well-sorted feldspathic medium sand occurs as a drape of the organic soil (Figure 2). A fossil trunk sending down roots in the organic soil layer yielded an age of 745 ± 47 cal. yr. BP (calibrated with a 2σ AMS age range of the fourth outermost tree ring, Supplementary Table S4-2). A 1.0 m thick unit of laminated silty sand is present above the unit of sand with trough cross-bedding. This laminated sand contains pristine valves of *Achnanthes bahusiensis*, *Cyclotella striata*, and *Hantzschia distinctepunctata* (Supplementary Figure S1). The unit of laminated silty sand is overlain by and interfingers with a 0.4 m thick unit of humic silty mud (Supplementary Figure S3C,D).

### 4.5. Alluvial Systems

The inland boundary of the dune field forms an irregular narrow zone (50–200 m in width) defined by the eastern terminations of alluvial fans composed of poorly sorted pebble-cobble gravels without a clay or silt matrix. The gravels, which form thicker deposits near the base of mountain slopes, are composed of weakly-metamorphosed pelitic rocks and radiolarian cherts (which are representative rocks forming the mountains).

## 5. Core Descriptions

### 5.1. Core Taken from Lake Obuchi

A core from Lake Obuchi was taken at 40°57′32.7′′ N, 141°21′07.0′′ E, and reached a total depth of 10 m. The lower half of the core mostly consists of well-sorted medium sand, whereas the upper half of the core consists mostly of mud (Figure 2A). The core was divided into eight stratigraphic units on the basis of lithology (Figure 2). These stratigraphic units are described as follows (from top to bottom):

Unit 1:  Unit 1 (1.5–0 cm core depth) is 1.5 cm thick silty ooze (Figure 3A) of algal remains and humic micro-debris, living and fossil foraminifers, very fine sand, and silt. The very fine sand and silt are composed of 20–30% quartz and 70–80% feldspars. The ooze yields the following diatoms: *Achnanthes biasolettiana*, *Achnanthes minutissima*, *Achnanthes lanceolata*, *Cymbella aspera*, *Cyclotella comta*, *Diploneis ovalis*, *Eunotia bilunaris*, *Eunotia implicata*, *Eunotia pectinalis* var.

minor, *Gomphonema parvulum*, *Hantzschia amphioxys*, *Navicula clementis*, *Navicula peregrine*, *Nitzschia parvula*, *Pinnularia karelica*, *Pinnularia viridis*, and *Stauroneis prominula* (Supplementary Figure S1). Calcareous benthic foraminifers are *Ammonia beccarii* (Linnaeus), *Buccella frigida* (Cushman), *Cibicides refulgens* Montfort, *Elphidium crispum* (Linnaeus), *Elphidium subarcticum* Cushman, and *Elphidium subincertum* Asano (Supplementary Figure S5, Supplementary Table S2). Agglutinated species do not occur. The ooze lacks sediment fabric.

Unit 2: Unit 2 (3.5 ± 0.5–1.5 cm core depth) is a 2-cm thick light-gray well-sorted fine sand (Figure 3A) composed of ~30% quartz, ~60% feldspars, and ~10% biogenic grains. The sand yields abundant calcareous foraminifers of *Ammonia beccarii* (Linnaeus) and *Cibicides refulgens* Montfort (Supplementary Table S2). Pristine frustules were not detected. The sand lacks bedding features and sediment size grading.

Unit 3: Unit 3 (9.5–3.5 ± 0.5 cm core depth) is a 6-cm thick gray well-sorted medium sand (Figure 3A) composed of 25–30% quartz, 55–60% feldspar, 5–7% heavy minerals, 2–4% rock fragments, and ~1% biogenic grains. The sand yields fragmented frustules of the following diatoms: *Achnanthes minutissima*, *Diploneis ovalis*, *Navicula clementis*, and *Pinnularia karelica* (Supplementary Figure S1). Calcareous benthic foraminifers are *Ammobaculites* sp., *Ammonia beccarii* (Linnaeus), *Cibicides refulgens* Montfort, *Elphidium crispum* (Linnaeus), *Elphidium subincertum* Asano, and *Trochammina hadai* Uchio (Supplementary Table S2). Agglutinated species of *Ammobaculites* sp., *Trochammina hadai* Uchio, and *Trochammina* sp. occur throughout the section. Broken bivalvian shells of *Macoma balthica* Yamamoto and Habe and *Potamocorbula amurensis* Schrenck occur from the upper half of the unit at core depth of 6.5 to 4.0 cm. The sand lacks bedding features and sediment size grading (Figure 3B).

Unit 4: Unit 4 (29–9.5 cm core depth) is a 19.5-cm thick dark-gray to dark-brown sandy organic mud (Figure 3A). Sand grains are composed of 30–35% quartz, 55–60% feldspar, ~1% heavy minerals, and 5–10% rock fragments. The mud contains abundantly pristine valves of the following diatoms: *Achnanthes minutissima*, *Cymbella aspera*, *Cyclotella comta*, *Diploneis ovalis*, *Eunotia implicata*, *Gomphonema parvulum*, *Navicula clementis*, *Navicula peregrine*, *Pinnularia karelica*, and *Stauroneis prominula* (Supplementary Figure S1). Calcareous benthic foraminifers are *Ammonia beccarii* (Linnaeus), *Cibicides refulgens* Montfort, *Elphidium crispum* (Linnaeus), *Elphidium subarcticum* Cushman, *Elphidium subincertum* Asano, and *Pararotalia nipponica* (Asano), and *Ammonia beccarii* occur throughout the section (Supplementary Table S2). Agglutinated species are *Ammobaculites* sp., *Trochammina hadai* Uchio, and *Trochammina* sp. The mud lacks bedding features.

Unit 5: Unit 5 (81.5–29.0 cm core depth) is a 52.5-cm thick gray medium/coarse sand (Figure 3A) composed of 25–30% quartz, 50–60% feldspar, 2–5% heavy minerals, 3–5% rock fragments, and 2–5% biogenic grains. The sand includes frustules of the following diatoms: *Achnanthes minutissima*, *Diploneis ovalis*, *Navicula clementis*, and *Pinnularia karelica* (Supplementary Figure S1). *Calcareous* benthic foraminifers of *Ammonia beccarii* (Linnaeus), *Cibicides refulgens* Montfort and *Elphidium crispum* (Linnaeus), and *Elphidium subarcticum* Cushman occur from the upper half of the section (29.0–48.0 cm; Supplementary Table S2). The sand yields planktic foraminifers of *Globigerinoides ruber* (d'Orbigny), *Globorotalia inflata* (d'Orbigny), and *Neogloboquadrina incompta* (Cifelli) in the middle of the unit (42.0–56.0 cm). Cross lamination is present in the top of the unit, and fining-upward cycles are present throughout the sand (Figure 3B). Well-rounded pebbles and shells occur at the transition from Unit 6 (80.0–81.5 cm).

Unit 6: Unit 6 (288–81.5 cm core depth) is a 206.5-cm thick dark-gray to black organic mud (Figure 3A). The unit contains beds of light-gray silty fine sand at two horizons (231.0–227.0 cm, 86.0–84.0 cm). The sand is composed of ~35% quartz and ~65% feldspars. A thin tephra layer is present at the top of this unit (289.0–288.0 cm core depth). The tephra is feldspathic and includes pyroxene and Fe-Ti oxide minerals. The pristine valves of the following diatoms are present throughout this unit: *Achnanthes minutissima*, *Cymbella aspera*, *Cyclotella comta*,

*Diploneis ovalis*, *Eunotia implicata*, *Gomphonema parvulum*, *Navicula clementis*, *Navicula peregrine*, *Pinnularia karelica*, and *Stauroneis prominula* (Supplementary Figure S1). The mud contains abundant *Ammonia beccarii* (Linnaeus) at three horizons (99.0 cm, 190 cm, and 251.0 cm) densely at three horizons (99.0 cm, 190.0 cm, and 251.0 cm). The unit lacks sediment fabric. An AMS age range of *A. beccarii* from the mud bed (82 to 83 cm core depth at the top of Unit 6 is 470 ± 49 cal. yr. BP (2 standard deviation)).

Unit 7: Unit 7 (414.0–288.0 cm core depth) is a 126 cm thick dark-gray tuffaceous mud (Figure 3A). The mud contains beds of light-gray fine sand at the depth of 361.0–359.0 cm. The sand is composed of quartz (~30%) and feldspar (~70%). *Ammonia beccarii* (Linnaeus) occurs abundantly at two horizons (99.0 cm, 190.0 cm, and 251.0 cm; Figure 1A). The mud contains pristine valves of the following diatoms: *Achnanthes minutissima*, *Cymbella aspera*, *Cyclotella comta*, *Diploneis ovalis*, *Eunotia implicata*, *Gomphonema parvulum*, *Navicula clementis*, *Navicula peregrine*, *Pinnularia karelica*, and *Stauroneis prominula* (Supplementary Figure S1). The mud lacks bedding features.

Unit 8: Unit 8 (10,000.0–414.0 cm core depth) is a 586-cm thick gray medium sand (Figure 3A) composed of 25–30% quartz, 55–60% feldspar, 6–9% heavy minerals, and 1–2% rock fragments. No fossils were observed. The sand lacks bedding and grading.

The Lake Obuchi core contains light-gray silty fine sand at four horizons: 426–414 cm (part of unit 8), 231–227 cm (part of unit 6), 86–84 cm (part of unit 6), and 12–11 cm (part of unit 4). In the Lake Obuchi core, laminae are visible in the organic-mud beds. Each lamination consists of algal remains and humic micro-debris in the upper part of a single lamina. The vertically sliced surface of dredged masses around the coring site exhibited a stack of fossiliferous medium sand (unit 3: 9.5–3.5 ± 0.5 cm) and overlying fine sand (unit 2: 3.5 ± 0.5–1.5 cm). Benthic foraminifera are ubiquitous in Unit 8 but not present in the lower units of sand and sandy mud. In Unit 8, two endemic bivalves, *Macoma balthica* Yamamoto and Habe and *Potamocorbula amurensis* Schrenck were found in situ. Figure 2 shows laboratory results from the cored sediments from beds shallower than 90 cm. Fossils are concentrated in some beds (Figure 2A). Most of argillaceous sediment lacks coarse grains (coarse silt + sand < 5 weight %), whereas most of the mud and organic matter contain very little sandy sediment (less than 2.5 weight %; Figure 2B). Mineralogical data from the samples from fining-upward sediment at 81.5–40.5 cm depth (Unit 5) are listed in Supplementary Table S1. Within this fining grain number exceeded 300 for each specimen. Quartz and feldspathic minerals account for 75% or more of all grains, and the remainder consists of heavy minerals, including orthopyroxene (enstatite), clinopyroxene (augite), magnetite, amphiboles, olivine, and biotite. Pyroxene amounts to approximately 75% of all the heavy minerals. The stratigraphic profiles of grain mineralogy are illustrated in Supplementary Figure S4.

The benthic foraminifera of 15 genera and 20 species were detected in the upper half of the Lake Obuchi core (from the core top to 414 cm depth), but are not present in the lower half of the core, between 450 cm and 1000 cm in depth. The benthic foraminiferal assemblages are dominated by calcareous species *Ammonia beccarii* (Linnaeus). In addition, *Buccella frigida* (Cushman), *Cibicides lobatulus* (Walker and Jacob), *Cibicides refulgens* Montfort, *Elphidium crispum* (Linnaeus), *Elphidium subarcticum* Cushman, *Elphidium subincertum* Asano, and *Rosalina bradyi* (Cushman) occur sporadically throughout the upper half of the core. *Ammobaculites* sp., *Haplophragmoides* sp., *Miliammina fusca* (Brady), and *Trochammina hadai* Uchio are rare, agglutinated species in the core. Scanning electron microscope (SEM) observation of the fossil tests shows the preservation of delicate surface ornamentation (Supplementary Figure S5). The sand-dominant beds at the depth interval between 81.5 and 1.5 cm (Unit 5 through Unit 2) particularly yielded abundant foraminifera. A list of fossils from these sediments is presented in Supplementary Table S2, and the depth profiles of fossil occurrence are shown in Figure 2C.

Benthic foraminifera submitted for the AMS age determination were collected from dark-gray mud (Unit 6). This dark gray mud of Unit 6 contains more than 5 weight % total organic carbon (TOC) (Figure 2C). The samples recovered from the beds of 82.5 and 249.5 cm core depth (Unit 6)

were used for both stable isotope measurement and age determination. The carbon isotope ratios of the foraminifera used for dating were −4.7 (82.5 cm) and −2.9‰ VPDB (249.5 cm; Supplementary Table S4-1), whereas the ratios of foraminifera used for the stable isotope analysis were −1.43 (82.5 cm) and −1.58‰ VPDB (249.5 cm; Supplementary Table S3-1). Furthermore, $\delta^{13}$C values are significantly different between them. Given that the light carbon ($^{12}$C) enrichment was caused by allochthonous organics remaining in the chamber (Supplementary Table S4-1), the heterogeneous residues might have restored the samples to old ages in contrast with their true ages. The carbon isotope ratios of the tests submitted for AMS age determinations (−2.4 to −5.91‰ VPDB; Supplementary Table S4-1) were one to three per-mil lower than those of foraminifera used for stable isotope measurements (−1.4 to −3.12‰ VPDB; Supplementary Table S3-1). The significant difference in the values implies that all of the AMS ages were older.

A negative shifting is recognized at core depths of 341.5, 237.5, and 41.5 cm, where the isotopic ratios fall below the theoretical calcite value in equilibrium with the lacustrine water in April (−3.82‰ VPDB; Supplementary Table S3-3). The heavy oxygen deficiency was likely caused by an extraordinary supply of thawing water (−10‰ VSMOW; [83]) into the lake. Contrarily, positive shifts are detected in the Lake Obuchi core at depths of 418.5, 399.5, 237.5, 133.5, and 3.5 cm. The isotope values increase to 0‰ VPDB, which is nearly equivalent to the value of modern planktic foraminifera living in the open sea off NE Japan (−0.5–1‰ VPDB; [84]) or theoretical calcite in equilibrium with the lacustrine water in November (−0.25‰ VPDB; Supplementary Table S3-3). Because individuals of *A. beccarii* do not grow in winter [85], the positive shifts in isotope values events are interpreted as being caused by large seawater influx into the benthic foraminiferal habitats. The oxygen isotope ratio on the remains of *A. beccarii* recovered from the deposit of the 2011 tsunami is –0.65 ‰ VPDB (Supplementary Table S3-1), which is 1.01‰ higher than the calcite theoretically equilibrated with the lacustrine bottom water (−1.66‰ VPDB) in June (temperature: 17 °C, salinity: 28 practical salinity units). The $\delta^{18}$O value of living *A. beccarii* collected from the lacustrine bottom surface on June 12, 2012, is −1.79‰ VPDB (Supplementary Table S3-2).

The carbon isotope data from Lake Obuchi show that the $\delta^{13}$C plots fluctuate around the line of 1.5‰ VPDB. The $\delta^{13}$C plots in Figure 2A show gradual fluctuations of carbon isotopes around the line of −1.5‰ VPDB, and the isotopic ratio of living *A. beccarii* (−2.69‰ VPDB; Supplementary Table S3-2) nearly establishes a limit on the negative side.

Accelerator mass spectrometry (AMS) age data from the tests of *A. beccarii* (Figure 2A), which represents an indicator benthic species of Lake Obuchi, are listed in Supplementary Table S4-1. At every depth interval, the sedimentation rate was calculated using radiocarbon dates and the age of the tephra.

*5.2. Core Taken from the Tashiro Wetland*

The Tashiro Wetland is located in Northeast Japan at an altitude of 570 m above sea level, at a distance of 45 km from the Lake Obuchi core site. The Tashiro Wetland has no inflow and runoff rivers. The core from the Tashiro Wetland was taken at 40°41′45.888″ N, 140°55′9.9186″ E using a thin-walled core sampler, and reached a total depth of 8.80 m. The core was divided into five stratigraphic units, on the basis of lithology (Figure 3B). These stratigraphic units are described (from top to bottom) as follows:

Unit 5: (781.0–0.0 cm) is homogeneous dark-gray peat with three felsic sand beds and three light-gray feldspathic tephra layers. The sand beds, approximately 3.0 cm thick, are intercalated at 551.0–548.0, 344.0–339.0, and 322.0–320.0 cm core depth. The tephra layers are present at 348.0–344.0, 76.0–65.0, and 62.0–58.0 cm core depth.

Unit 4: Unit 4 (798.0–781.0 cm) is 17.0-cm thick dark-gray clayey mud.

Unit 3: Unit 3 (820.0–798.0 cm) is 22.0-cm thick dark-gray massive peat with volcanic pebbles.

Unit 2: Unit 2 (864.0–820.0 cm) consists of 21.0-cm thick coarse sand with thin felsic tephra at the core depth of 842.0–839.0 cm.

Unit 1:　Unit 1 (880.0–864.0 cm) is 31.0-cm thick light-gray pumiceous tephra.

Within Units 1–5, 81 genera were identified that were assignable to a formally descriptive fossil classification (e.g., [56]). The palynological results on arboreal (43 genera) and herbaceous (34 genera) pollen are listed in Supplementary Table S5 as a pollen diagram of 790 samples, which is expressed as individual percentages relative to the total pollen in each slide. Among them, *Abies*, *Picea*, *Tsuga*, *Pinus*, *Haploxylon*, *Diploxylon*, *Betula*, *Quercus*, *Fagus*, and *Cryptomeria* showed sharp fluctuations in occurrence. Optical photomicrographs of representative fossil pollen are shown in Supplementary Figure S6.

## 6. Interpretation of the Outcrops/Exposures and the Cores

### 6.1. Outcrops and Exposures among the Coastal Aeolian Dunes

In the outcrop of Tatenuma, the presence of gravel suggests bedload transport (e.g., [86]) and deposition as channel gravels in a point bar (e.g., [87]). This sediment succession implies the growth of a point bar sequence under diminishing water flow (e.g., [88]). Abrupt channel abandonment likely occurred during deposition of the silty mud. The sedimentary characteristics of this gravelly sand imply deposition by water with large transport capacity, possibly in a stream or river channel (e.g., [89]). The sand-gravel sequence preserved in Paleosol IV represents the transition of a fluvial system from straight through meandering to abandoned channels. The channel abandonment is reflective of the cutoff of the watercourse in the process of meandering. The sheet-like deposition of silty fine sand in the humic mud of Paleosol IV was the product of overbank flows into the surrounding low-lying areas (e.g., [90]), and the shift from alluvial to flood plains might have occurred after the filling of disconnected waterways with muddy sediment. Decreased water supply from the upstream regions and diminished sediment influx promoted vegetation that covered poorly drained and flat bottomlands (e.g., [91]). Thus, it is conceivable that the sequence exhibits the details of a changing waterway during the formation of Paleosol IV. Figure 5 shows the interpreted geological model explaining the evolution of a drainage system in the dune field. The geological reconstruction clarifies that the size of a straight channel exceeded 10 m in width and 2–2.5 m in water depth. Dune III stopped growing at ~500 years cal. yr. BP [40]. Afterward, the dune field underwent an extensive development of soils (Paleosol IV). The fluvial reconstruction and the age calibration imply the short-lived emersion of a large-scale fluvial system during the early 17th century.

The imbricated gravels and trough cross-bedded sands in the stratigraphic unit of Paleosol IV (Supplementary Figure S3B), which formed during the early 17th century (Figure 5), reflect large torrential currents with massive flows. Such terrestrial water release in the dune field of Shimokita might have occurred via groundwater flow. The reconstructed climate mode during the early 17th century suggests that the weather conditions at that time were warm and rainy/snowy (Figure 6). Significant draining likely continued in the dune field for several decades. There is no modern river in the Shimokita Peninsula of comparable scale and geometry to the reconstructed channel, and any steady outflow evidence is absent in modern estuaries. Thus, the water supply in the upstream areas was likely fairly large during that time, and weather conditions appear to have been rainier than in modern conditions. The Shimokita Peninsula was likely susceptible to the stronger influence of monsoonal circulation. The full-scale outcrop disclosing a well-developed fluvial-alluvial sequence is the only observation made in the area. However, restricted outcrops of sand (interpreted as rapid water deposits) are scattered through the dune field. Each time a humid climate prevailed in NE Japan, short-lived fluvial conditions likely emerged in the area.

Sediment liquefaction and dewatering were accompanied by gravitational sliding of the fluvial channel walls. The stream damming by levee collapse led to channel abandonment. The AMS age data on tree rings from flood plain deposits (Supplementary Table S4-2) imply the fluvial system ended during the 17th century (Supplementary Figure S3). Given the geological observations in the field of the sand bed deposited by the surging flows of a tsunami nearly 700 years ago [40], the lower sand without obvious sedimentary structures and the upper laminated sand exposed along the fluvial

valleys (described in Section 4.4) correlate chronologically with the deposits of the historical tsunami. Surging currents of the tsunami are thought to have coursed upstream through rivers. The abrupt filling of the watercourse with huge amounts of sediment reduced the drainage capacity of the fluvial systems, probably leading to the reduction of topographic fall.

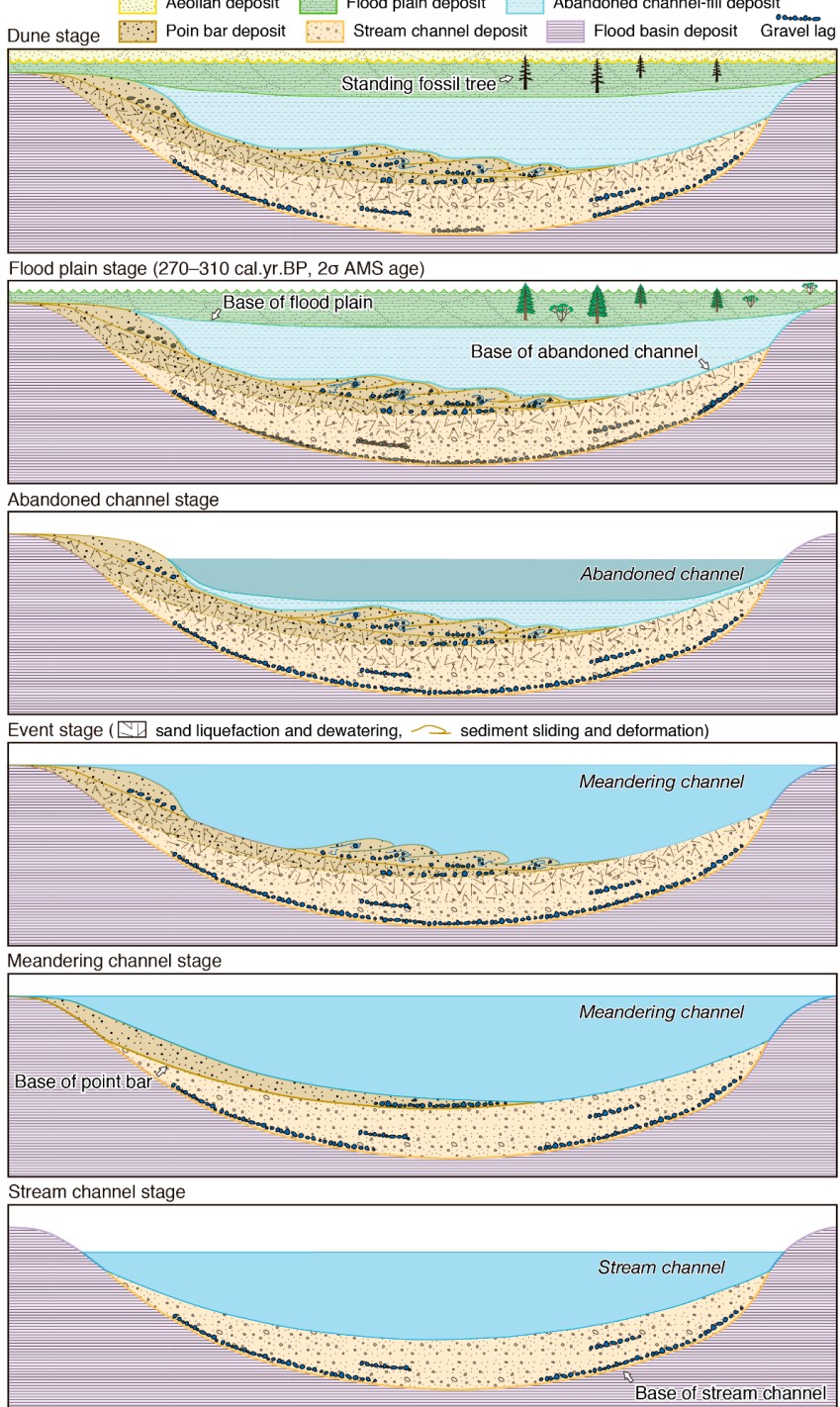

**Figure 5.** Six-stage geological process-respondent diagrams illustrating the evolution of a fluvial-alluvial system preserved in the flood basin sequence of Paleosol IV. The genesis of three different channels, stream, meandering, and abandoned, typifies the fluvial evolution. The channel processes were interpreted by detailed observation and analysis of the vertical profile shown in Supplementary Figure S3. The site of the geological exposure is shown in Figure 1D.

The homogenized nature of the massive sand with mud flakes is interpreted as the product of sediment liquefaction or fluidization, which have various causes, such as overloading [92], seismic activity [93], rapid deposition [94], tsunami [95], and storm waves [96]. In the lower unit of sand without obvious sedimentary structures (described in Section 4.4), the sand dike (within the upper unit of laminated sand) penetration into the overlying penecontemporaneously deformed and broken silty fine sand bed implies the outbreak of severe shaking (and consequent liquefaction), probably by earthquakes. The plastic deformation of mud in point-bar deposit suggests the occurrence of gravitational river-slope failure before the channel cutoff (Figure 5). The mineralogical (feldspars and quartz) and sedimentological (well-sorted and coarse-grained sand) properties of injection material in the slumped bed (Supplementary Figure S3D) are common for stream channel sand, which implies that the liquidized part of the underlying layer is the source of the injected sediments. Failure of the slope may have occurred simultaneously with sediment homogenization by bed shaking. The inferred river slope failure and the channel cutoff likely reflect the extent of earthquake damages to coastal aeolian dunes. Such collapse of embankments and the consequent damming of rivers with sediment were observed at the time of the 2011 Great Earthquake (moment magnitude (Mw) = 9; [40]). Likewise, the 1983 Japan Sea earthquake (Mw = 7.7–7.8) caused intensive shocks in the coastal zones, and the seismic vibration triggered the collapse of aeolian dunes and riverbanks [97].

### 6.2. Outcrops and Exposures in the Fluvial Environments

In exposures described in Section 4.4 (exposures along the river with its source at Lake Tatenuma), the well-sorted and feldspathic medium sand overlying the humic soil of Paleosol III is correlative to the overlying parallel-bedded sand of Dune III. The similarity of these two units implies that the unit of sand without obvious sedimentary structures and the unit of laminated silty sand were deposited just after the deposition of aeolian sediment. The truncation of standing fossil trunks by a bed of sand with parallel-lamination also suggests that the arboreal death occurred by burial in aeolian sands. Several other features in exposures described in Section 4.4 (exposures along River Zaimokuzawa) are suggestive of strong currents. The mud flakes (interpreted as rip-up clasts) within the basal part of the lower unit in the sand suggests that inrushing rapid flows went up through river valleys and swept up pieces of the muddy substrate. The gray and black laminae that resemble inverse grading (described in Section 4.4) probably arose from the repetitive generation of grain-flow-like traction carpets [98], and the laminated sand is interpreted as deposition under supercritical flow conditions.

### 6.3. Lake Obuchi Core

Several tephra beds are present in Unit 7 of the Lake Obuchi core. Comparing the chemistry of the tephra beds from the Lake Obuchi core with previously published data on the chemistry of tephra beds from the Towada volcano (located at 20 km south from the Tashiro Wetland; Figure 1B) in the Aomori Prefecture [99] indicates that the Lake Obuchi tephra is characteristic of a tephra from the Aomori prefecture named the Towada-B tephra [99], which has been dated at ~1.90 thousand years (kyr); [100,101]. On the basis of tephra correlation and assumptions of sediment accumulation rates, it is possible to construct a plot of age versus depth for the Lake Obushi core (Figure 3A).

The rate of organic matter accumulation in the Lake Obuchi depends on seasonal productivity [82], which is generally controlled by nutrient supply and light availability (e.g., [102]). Both factors increase under wet and warm conditions in the summer season. The scarcity of abraded quartz grains in the muddy beds is interpreted to represent declines in aeolian sediment mobilization during the intervals of mud accumulation. The occurrence of enstatite and augite from the tuffaceous mud (Unit 7 of the Lake Obuchi core; 414–288 cm core depth) suggests that the tuffaceous mud was derived from Miocene volcanic rocks. Given that the terrestrial development of soils advances with the progress of extensive chemical/biochemical weathering, it is possible that thicker mud beds were formed under a longer continuation of warm-wet climates.

The abrupt appearance of coarse sand over organic mud at 81.5 cm in core depth (transition from Unit 6 to Unit 5; Figure 2A) implies an abrupt change in lake conditions from stagnant to active waters at around 400 yr. BP. The concentration of well-rounded pebbles and shells (see Supplementary Figure S3A) at the transition likely suggests the sudden extension of seawater torrents to the lake. It is probable that surging currents opened a gateway and reached the coast. The succession of maximum peaks that appear in the stack of output profiles of grain size was divided into lower coarse (c), middle intermediate (i), and upper fine (f) columns (Figure 2B). Each succession was likely established under the respective kinetic control of water flows, such as settling of finer grains under weaker currents. The common heavy-mineral constitution in the Lake Obuchi core and in modern beach sand suggests that the seashore was the source of the sediments in the core. As waves surged toward the coast, marine sediments were probably swept up and transported into the lake through the gateway. The planktic foraminifera *Globigerinoides ruber* (d'Orbigny), *Globorotalia inflata* (d'Orbigny), and *Neogloboquadrina incompta* (Cifelli) occur from the middle and upper columns (56–42 cm; Supplementary Table S2, Figure 2C). *Neogloboquadrina incompta*, however, is the index species of planktic foraminifera characteristic of the Tsushima Warm Current [103]. Therefore, the upward-fining sediments that contain *Neogloboquadrina incompta* (Unit 6 of the Lake Obuchi core) are interpreted as having been deposited by water currents that originated offshore, and entered the Lake Obuchi via run-up currents of the tsunami that occurred during the 17th century.

The preferential occurrence of *A. beccarii* in organic matter-rich deposits, as compared with other benthos (Figure 3A), is ascribed to the high physical capacity against an unfavorable state of living. *Ammonia beccarii* lives in brackish water [85] and is adaptable to high-organic and low-oxygen habitats [104]. Mud is the main lithology in the upper half of the core (Unit 1 through Unit 6), whereas the lower half (Unit 6 through 8) mostly consists of well-sorted medium sand (Figure 2A). The occurrence of oyster reefs from the transitional horizon (Unit 8, at 470 and 450 cm in depth; Figure 3A) denotes the shift of lacustrine environments from a shallow open lagoon to a coastal lake. Thus, it is likely that oxygen-deficient conditions have prevailed in the lake since its isolation from the sea. Broken pieces of the bivalves *Macoma balthica* and *Potamocorbula amurensis* occur throughout the upper half of the core, suggesting that the present molluscan habitat is traceable back to the onset of the mud supply (Unit 7 at 414 cm in depth).

*Achnanthes bahusiensis*, *Cyclotella striata*, and *Hantzschia distinctepunctata* are representative marine species in NE Japan [105], and their presence implies the occurrence of a marine invasion with vigorous flows of high grain concentration. The thin sand bed at 3.5 ± 0.5–1.5 cm (unit 2) is interpreted as a suspension deposit that took place when seawater mixed with lake water. Nemoto et al. [82] reported the occurrence of fine sand from the subsurface of bottom deposits recovered by an Ekman-Berge sampler after this tsunami event.

*A. beccarii* is thought to have maintained its existence in Lake Obuchi in the salty conditions resulting from the incursion of marine waters by the tsunami event. The $\delta^{18}$O positive excursion equivalent to the level of rise in the isotope ratio seen on the tests from the 2011 event bed is noticeable at the subordinate horizons of the Lake Obuchi core at depths of 40.5 cm (Unit 5), 132.5 cm (Unit 6), 248.5 cm (Unit 6), and 397.5 cm (Unit 7) (Figure 2A). Although the presence of sand grains was not recognized at every bed, the $^{18}$O enrichment in excess of 1‰ implies that large seawater influxes caused each $\delta^{18}$O short anomaly. Application of the age model on the Lake Obuchi core suggests that the age of the isotope anomaly that is present at ~132.5 cm is estimated at 965 ± 5 cal. yr. BP.

As the tsunami invaded the coastal zone, Dune III started to grow under increasing mobilization of aeolian sediment (Supplementary Figure S2). Sediment beds deposited by the 2011 earthquake tsunami were easily removed through subsequent erosion and weathering [106]. Thus, the immediate burial by wind-driven sand is likely the cause of the preservation of the tsunami deposits [107].

At ~81.5 cm core depth, the abrupt appearance of a bed of sand (base of Unit 5) over organic mud (top of Unit 6) is interpreted as the result of an abrupt change in lacustrine conditions from stagnant water to active flows (Figure 2A). The influx of seawater is suggested by the occurrence of marine

mollusks from the base of this sand and by the occurrence of planktonic foraminifera in the middle of this sand.

*6.4. The Tashiro Wetland Core*

The Tashio Wetland core contains a continuous and detailed record vegetation changes. Faint laminations, along with a lack of plant roots, reflect the absence of bioturbation throughout the core. The one tephra bed in Unit 1 and the three tephra beds within Unit 5 of the Tashiro Wetland core were identified mineralogically and petrochemically as being correlative with known tephra deposits from the Towada volcano, which is located just 30 km south from the Tashiro Wetland. Specifically, the tephra bed at 58–62 cm depth (Unit 5) in the Tashiro Wetland core is correlated with Baekdu-san-Tomakomai (B-Tm: 62–58 cm, 985 ± 20 cal. yr. BP; [108]). The tephra bed at 65–76 cm depth (Unit 5) in the Tashiro Wetland core is correlated with Towada-A (To-A: 76–65 cm, 990 ± 195 cal. yr. BP; [102]). The tephra bed at 338–347 cm depth (Unit 5) in the Tashiro Wetland core is correlated with Towada-Chuseri (To-Cu: 347–338 cm, 6246 ± 67 cal. yr. BP; [109]). The tephra bed at 880–849 cm depth (Unit 1) in the Tashiro Wetland core is correlated with Towada-Hachinohe (To-H: 880–849 cm, 15875 ± 315 cal. yr. BP; [55]).

Together with the well-preserved textures, the lack of secondary carbonate precipitation or shell browning proves that the fossil shells did not undergo chemical/biochemical reforming after burial. The results of the stable isotope analysis on fossil and living *A. beccarii* are listed in Supplementary Table S3-1 and S3-2, respectively. The analytical data on fossil and living foraminifera sampled at the coring site, as well as the isotopic results of lake water, are displayed in Figure 2A. No definite stratigraphic correspondence was noted between the oxygen and carbon isotopic fluctuations. In view of the high-organic and oxygen-deficient habitat during the growth stage [82], it is likely that the shell carbonates were formed under the control of complicated carbon circulation in the lake. The isotopic information on lacustrine waters is indispensable for evaluating the past limnological conditions from the perspective of biomineral isotopes.

## 7. Discussion

*7.1. Climatic Influence on Dune Growth*

Averaged air mass movement is estimated roughly on the basis of atmospheric water circulation, and oxygen isotopes of seasonal precipitation are general indicators of hemispheric air dynamics. Figure 6 shows the distribution of $\delta^{18}O$ (‰ VSMOW) during summer (May–September) and winter (November–March) precipitation ([110]; mean values during 1960–2001). The approximate $\delta^{18}O$ values of summer and winter precipitation in Siberia are −10‰ and −25‰ VSMOW, respectively. During summer, $^{18}O$-enriched rainwater falls in Siberia, and the Southeast (SE) Asian monsoon transports warm and wet air to the interior of Siberia. The trend of the summer monsoon is strongly controlled by low-latitude ocean circulation and, to a lesser extent, the prevailing westerly winds [111]. The Siberian monsoon moves the winter polar air as far south as the Philippines. Under this situation, the Siberian high-pressure system supplies atmospheric moisture, highly depleted in $^{18}O$, to the south and east. However, the oxygen isotope values of winter precipitation in NE Japan are the same, being as high as −10‰ VSMOW (Figure 6). This isotope discrepancy arises from the input of heavy oxygen into the atmosphere during air mass migration, and the meteorological concept of air mass moistening over the Japan Sea is supported by the isotope data from seasonal precipitation.

The distribution of oxygen isotopes clearly shows the dynamics of each monsoonal territory (expansion or retreat) responsible for seasonal changes in East Asia. Northeast Japan is located where the winter and summer monsoons extend directly (e.g., [54]). Thus, it is possible that the meteorological events originating from sudden changes in atmospheric circulation may never go far from the dune field. The annual precipitation in the Shimokita Peninsula (1013 mm) is lower than the national average in Japan (1757 mm) according to the 1981 to 2012 records (The Meteorological Agency of Japan, www.jma.go.jp/). The Shimokita Peninsula dune field is located in an area of high snowfall, and nearly

half the annual precipitation is in the form of snow (average 569.8 mm; The Meteorological Agency of Japan, www.jma.go.jp/). The snow thaw period is short in the Shimokita Peninsula relative to that in the Japan Seaside, where it extends to late April.

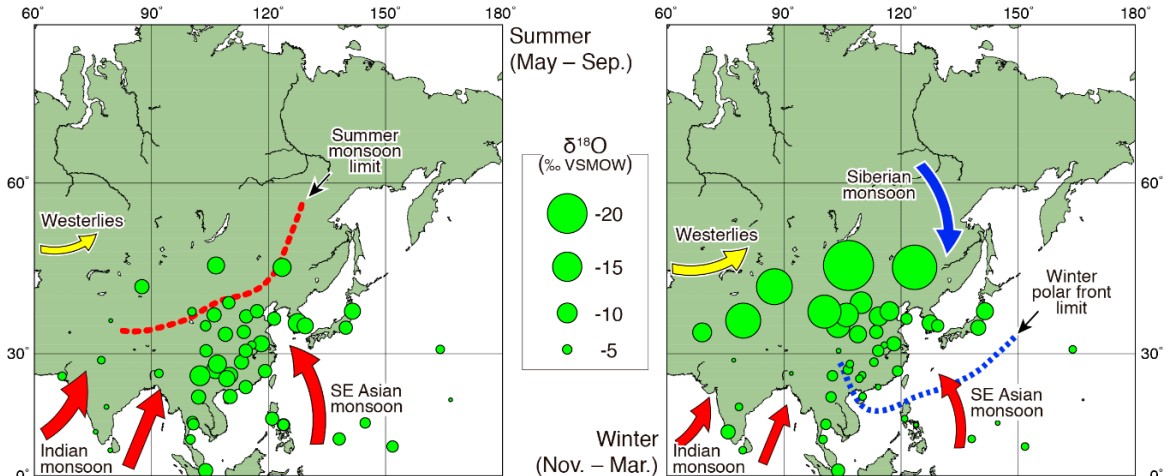

**Figure 6.** Distribution of $\delta^{18}O$ (‰ Vienna Standard Mean Ocean Water (VSMOW)) in summer (May–September) and winter (November–March) precipitation. The oxygen isotopes are the mean values through the interval between 1960 and 2001 (International Atomic Energy Agency data, https://nucleus.iaea.org/wiser/index.aspx). The present-day average limits of the summer monsoons and the winter polar front are based on results reported by [112–115]. The polar front extends to the north of the Philippines in winter seasons, whereas the summer monsoon reaches the east of Lake Baikal. The hemispheric-scale shifts in atmospheric circulation suggest the possible occurrence of torrential rains and heavy snowfall in NE Japan under the increased air-mass humidity over the Japan Sea. The color-cording shows the climate zonation through the last 15.5 kyr.

### 7.2. Holocene Paleoclimate Reconstruction

A reconstructed time series of Holocene climate parameters are illustrated as thin black lines in Figure 7, which explains the detailed transition of annual mean temperature and annual precipitation during the last 15.5 kyr. The profiles reflect the post-Last Glacial Maximum (LGM) atmospheric evolution in northernmost NE Japan. In the profiles, two major atmospheric events occur, which are likely related to the Holocene Climate Optimum (~6 kyr BP; [112]; https://www.ncdc.noaa.gov/global-warming/mid-holocene-warm-period) and the Tsushima Current influx into the Japan Sea (8.3 kyr BP; [116]). This current began to stream into the Japan Sea at ~8.3 kyr BP [116]. This oceanic event triggered increased evaporation of the sea surface [40], and the growth of snow clouds likely intensified along the eastern margin of the Japan Sea from this time forward. Thus, fluctuation of the prevailing westerlies might have been involved in the climatic evolution of NE Japan [117]. These profiles (Figure 7) also include four climatic events in the Northern Hemisphere [118], namely the Bølling oscillation (14,650–14,300 years BP), the Allerød oscillation (14,080–13,610 years BP), the Younger Dryas (12,850–11,650 years BP), and the Pre-Boreal (11,650–10,500 years BP). Conditions during the Holocene climate optimum (~6 kyr BP) in this region probably reached atmospheric conditions that were ~2 °C warmer and ~200 mm dryer compared with those of the preceding early Holocene. After the Holocene climate optimum (~6 kyr BP), the annual mean temperature and the total precipitation returned to the present levels, with exceptional cases of ~1 °C warmer and ~50 mm dryer conditions at ~2 kyr BP.

The series of three-point running averages of the calculated values shows the general trend of increase or decrease in air temperature and precipitation with an accuracy of 100 years or less (Figure 7; red and blue thick lines). In the case of two independent variables fluctuating with the lapse of time, four modes of combination are shown in the patterns of chronological changes. Referring to the annual mean air temperature and the annual precipitation, the following four climate models are

apparent; (1) increase in temperature and decrease in precipitation; (2) increases in both temperature and precipitation; (3) decreases in both temperature and precipitation; and (4) decrease in temperature and increase in precipitation. These climate modes may be used to categorize the past details of air temperature and annual precipitation during the past 15.5 kyr (Figure 7).

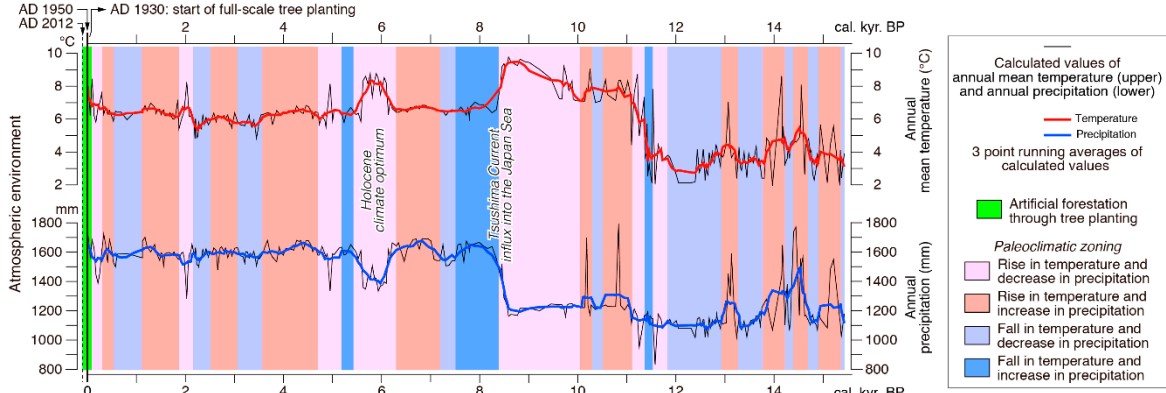

**Figure 7.** Reconstructed annual mean temperature and annual precipitation during the last 15.5 thousand years (kyr). (black lines) based on pollen spectra and best modern analog (BMA) approaches. The red and blue lines represent three-point running averages of the calculated values of temperature and precipitation. The four climatic zones provided in this figure are based on the combination of increases and decreases relative to the chronological movement appearing in the averaged profiles. The rise in both temperature and precipitation after 70 years BP is false. It is likely that the assemblages of airborne pollen have rapidly changed as a result of afforested area expansion inside the basin since the late 19th century (full-scale afforestation began in AD 1930). The pinaceous pollen increase likely leads to the unfavorable results. The notable atmospheric events regarding the Northern Hemisphere climate change [118] are manifested in the profiles. The Tsushima Warm Current intrusion into the Japan Sea has resulted in intensified precipitation in NE Japan.

### 7.3. Atmospheric Background on Local Vegetation

Because the weather conditions of the Shimokita Peninsula are generally the same as those of the Tashiro Wetland ([70]; The Meteorological Agency of Japan, www.jma.go.jp/), the obtained results from the Tashiro Wetland core may be applied to the reconstruction of the paleo-climate in the Shimokita Peninsula. The paleoclimate transition (Figure 7) shows that the last stage of cool and dry mode began ~1050 years ago and continued until ~550 years ago. The last cool/dry climate stage in NE Japan was followed by a phase of warm and humid weather that began at around 550 cal. yr. BP and lasted for about ~200 years. The climatic warming in NE Japan agrees well with the evidence for continental warming deduced from the magnetic susceptibility of lake deposits in China [119]. Climate proxies deduced from the geological data on loess and lacustrine sediments in China indicate that the winter monsoon intensified during approximately the 10th century and continued as late as the Little Ice Age [119–121]. Historical documents on the meteorological conditions in China [122] suggest that the onset of cooling was likely caused by cold air masses originating in Siberia [123]. The synchronistic changes in climate between NE Japan and China, together with the global atmospheric events, suggest that the paleoclimatic reconstruction and zoning has hemispheric application.

The dominance of piney pollen in recent years (Supplementary Table S5) is thought to be the influence of afforestation (forest growth and expansion) around the Tashiro Wetland. Thus, the abrupt increase in temperature and precipitation after 70 yr BP is not natural and was likely caused by extensive afforestation. It is probable in the opposite sense that the numerical estimate of climatic warming supports the validity of paleoclimate reconstruction on the basis of pollen records. Around the same time, the planting of pinaceous trees was practiced in the Shimokita Peninsula to prevent aeolian grain motion [45]. Full-scale afforestation began in AD 1930 throughout the dune field [50].

Before the planting, the area was largely devoid of vegetational cover, and the dunes were exposed to wind erosion [51].

The pollen chronostratigraphy from the Tashiro Wetland core reveals the occurrence of a major shift in the terrestrial vegetation ~8500 years ago. *Quercus* decreased to ~15% and *Fagus* increased to 50% or more (Supplementary Table S5). These palynological data are consistent with previously published paleobotanical data [124], which concluded that the arboreal flora in northern NE Japan largely changed ~8500 years ago and that *Fagus* spread rapidly afterward. The full-scale influx of the Tsushima Current into the Japan Sea began at 8.3 cal. kyr BP [116]. Afterward, the warm northern current likely exerted atmospheric humidity over the sea and these wetter air masses might have brought heavy snowfalls during the winter to NE Japan [125]. Cool temperate broadleaved forests dominated by *Fagus* began to expand in northernmost NE Japan at ~8 ka [70]. The rapid expansion of beech forests implies atmospheric humid increases triggered by the warm current influx. Thus, the floral shift was the outcome of global change in the context of eustatic control on local climates. Between 6.3 and 5.7 cal. kyr BP, a marked increase in abundance of *Alnus* occurred simultaneously with a reduction of *Fagus* (Supplementary Table S5). This floral change reflects a temporal decrease in the amount of precipitation (~200 mm decrease in annual precipitation), which implies climatic aridification during this time (Figure 5).

### 7.4. Meteorological Impact on Dunes

The climate mode of high temperature and high precipitation, including the 1–1.5 °C increase in temperature and 50–100 mm increase in annual precipitation relative to the previous climate stage (Figure 7), is reflected by the major occurrence of *Fagus*. The palynological results suggest that the stratigraphic appearance of *Fagus* is an index of high snowfall, especially snowfall enhanced in winter seasons during periods of higher temperature. Under conditions of climate warming, the dune field likely experienced frequent snowmelt floods during the spring thaw, whereas during pluvial seasons, torrential rains might have triggered repeated flooding. The botanical affinities of fossil taxa in northernmost Tohoku fell into warm and cool/cold groups [70]. Sub-boreal coniferous and broad-leaved species (e.g., *Abies*, *Picea*, *Tsuga*, *Pinus*, and *Betula*) are cool/cold climate indicators, whereas cool to temperate broad-leaved species (e.g., *Fagus*, *Quercus*, and *Carpinus–Ostrya*) are representative of warm-climate conditions.

### 7.5. Global and Local Control on Dune Growth

Figure 8A,B show the chronostratigraphic profiles of the lacustrine and terrestrial deposits through the middle and late Holocene. The reconstructed isotope chronostratigraphy is reflective of the changing coastal water system in the Shimokita Peninsula through the last 4000 years (see Figure 8A). Evidence of terrestrial weathering represented by the formation of soil layers is not present in sediments/strata older than 4 kyr BP. The climate zonation in Figure 8C displays the periodical fluctuation in atmospheric conditions during the last 6 kyr. Dune growth began at ~6 kyr BP under a warm and dry climate (Figure 8D). The reconstructed profile of global sea-level fluctuation in Figure 8E shows the details of coastal retreat and advance after the post-LGM [126]. During the middle Holocene sea-level highstand of ~6 kyr [3,5,6], wave-induced longshore currents probably reworked and redistributed sediments on the shoreface [127]. The chronological correlation of the aeolian sediments (Dune I to IV; Figure 8D) with sea-level changes (Figure 8E) demonstrates that the dune growth corresponded with the global regression, with aeolian development occurring on a larger scale under conditions of sea level fall of longer durations. Seaward shoreline migration was probably responsible for the episode of sediment reworking, and inshore currents and waves are thought to have transported sand-sized grains shore from lowstand deposits. Dune I, the largest dune unit, was likely produced by continuous aeolian sediment mobilization during low sea-level conditions for at least 2000 years. Because Holocene tectonic movement has been negligible in Shimokita [64], the eustatic regression during the period of post-glacial highstand was the major cause for the dune growth on a massive scale along the Pacific

coast of Shimokita. The aeolian growth of Dune II began about 300 years after the onset of regression. Although the reason for this time lag is unknown, the repeated occurrence of fossil forests during this time (Figure 8B) suggests that the vegetation might have influenced aeolian grain transport.

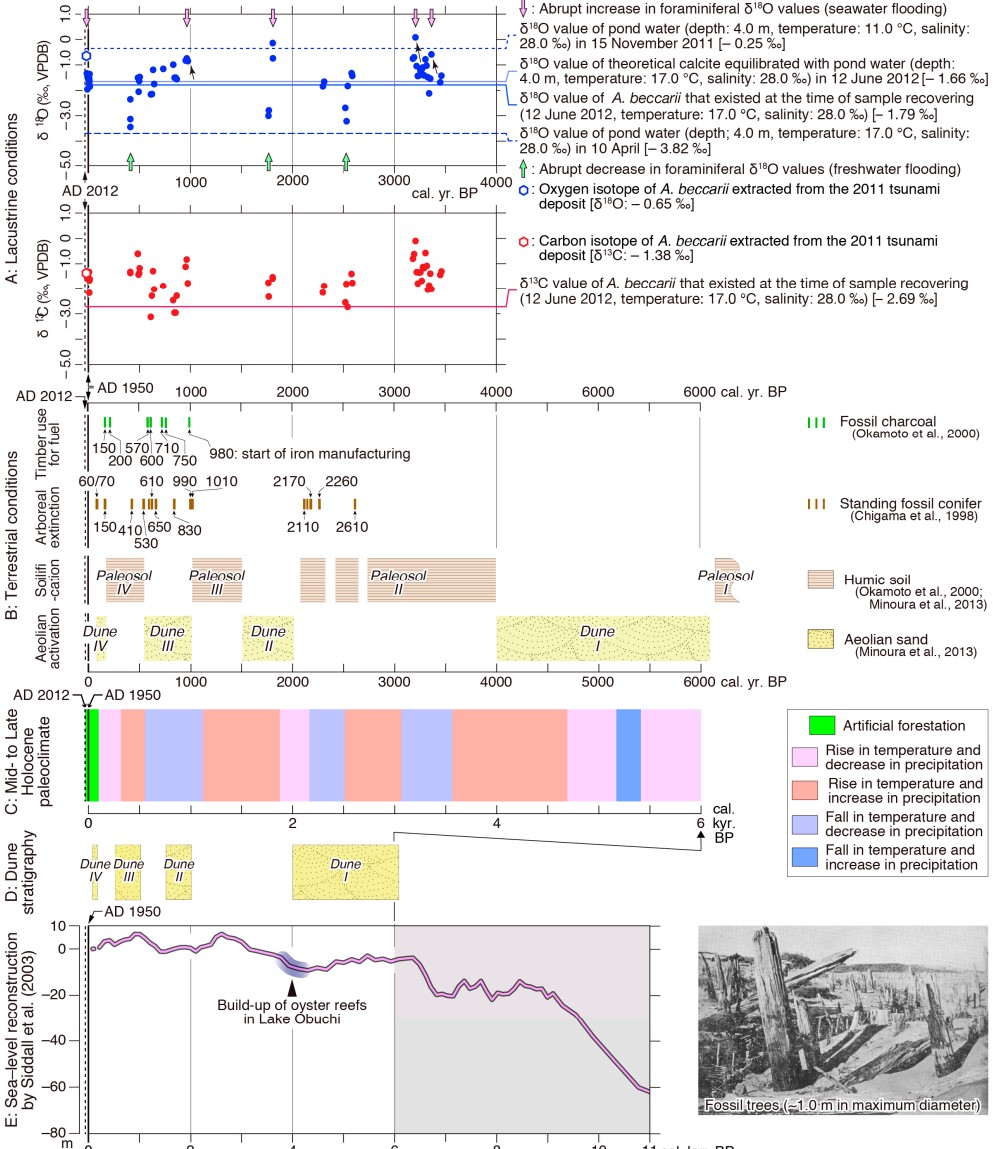

**Figure 8.** Chronological correlation of changes in lacustrine and terrestrial conditions, paleoclimate, and sea-level. (**A**) benthic foraminiferal oxygen and carbon isotopes; (**B**) terrestrial environments, including, from top to bottom, timber felling, forest extinction, solidification, and aeolian activation; (**C**) paleoclimate. The terrestrial evolution is indicated by the chronological transition of aeolian activation (Dunes I to IV; [40]), surface solidification (Paleosols I to III; [50]), and arboreal extinction [44]. The climate zoning is derived from Figure 5. The (**D**) cross-correlation of the coastal dune chronostratigraphy [40] with the (**E**) sea-level reconstruction [126] reveals aeolian activation corresponding to global regression. The build-up of oyster reefs in Lake Obuchi implies the temporal duration of intertidal lagoons with the transgression that began about 4.5 kyr BP. The charcoal fossils are the remains of timber burning for iron manufacturing [45]. The forest death was estimated by the age data of standing fossil conifers. The subsurface shell samples of *A. beccarii* are from the deposits of the 2011 tsunami. The photo in the lower right was taken about 100 years ago, showing the occurrence of fossil forests lying buried in sand of Dune III (modified from Minoura et al. [51]). The exposure of fossil trees was caused by the aeolian activation in the stage of Dune IV.

Hemispheric warming has probably been a major factor in modifying the dune landforms through surface erosion and weathering in Shimokita since 8.3 kyr BP, and the dune landforms reflect the integrated effects of climate change on a hemispheric scale. The Holocene paleoclimate model shows the frequent prevalence of climatic warming and the consequential high rainfall that has occurred since 6 kyr BP. On such occasions, topographic leveling caused by erosion and weathering is presumed to have extended throughout the Pacific coast of the Shimokita Peninsula. The more restricted distribution of older dunes might be a geological reality (Figure 1D), suggesting the repetition of deep denudation in the dune field.

The foraminiferal $\delta^{18}$O records in Figure 8A provide evidence of freshwater flooding at the three horizons of the Obuchi core, with the age of the last being calibrated to ~400 years ago. The chronological correlation of the isotope profile with the reconstructed paleoclimate transition (Figure 8C) reveals that each flood occurred during an interval of climatic warming. The enhanced influx of the Tsushima warm currents into the Japan Sea promoted the humidification of westerly air masses owing to accelerated evaporation of the sea surface, which led to the growth of snow clouds along the island arc [61,125]. The rising trend of atmospheric temperature stimulated the northward penetration of summer monsoons (Figure 7). Thus, well-developed rain clouds could have easily reached northernmost Japan during those time intervals. The historical archive includes a severe flood disaster in Shimokita that was caused by torrential rainfall in 1578 AD [44]. Following the paleoclimate reconstruction, it is likely that the warming over 0.5 °C increased the potential of heavy precipitation in NE Japan. The AMS dating of a fossil tree from this deposit indicates the appearance of a river with large flows at about 400 years BP (Figure 5), which is consistent with the age of the freshwater flooding in the lake (Figure 8A). The thick point bar sequence (Supplementary Figure S3A) implies a massive sediment discharge through fluvial channels. Frequently-occurring heavy rains and snowfall likely maintained the large streams in the area within a limited span of several decades. Arboreal forests were relatively sparse during warm and wet climates (Figure 8B) and the dune landforms were dissected through surface weathering/erosion under conditions of reduced aeolian sediment mobility. The extensive development of Paleosol III (Figure 4) suggests a stable chemical/biochemical weathering and the consequent deep erosion of the underlying layers (Figure 8B). The paleoclimate model reveals the control of persistent temperate-humid conditions during this time interval (Figure 7).

*7.6. Effects of Marine Events on Aeolian Dune Landforms and Coastal Environments*

Large-scale tsunamis likely invaded the coastal zone at least five times since 4 kyr BP. By the application of the age model on the Obuchi core, the date of the isotope anomaly from the horizon of about 132.5 cm is estimated at 965 ± 5 cal. yr. BP. In view of the sediment deposition from rapid flows with high grain concentration and the dominant occurrence of saltwater frustules, it seems likely that the Pacific coast of Shimokita experienced a tsunami during the 12th century. This age is approximately 200 years younger than that of the seawater flooding suggested by the dated isotope anomaly. No ancient document exists on Shimokita tsunami disasters that occurred between the 10th and 11th centuries [128]. Thus, the isotope event detected at about 132.5 cm in depth is associated with the tsunami invasion of the 12th century. In the previous section, we considered the origin of the sand rich in mafic grains and magnetite to be related to the deposition produced by the tsunami run-up. The 2σ AMS age range of 745 ± 47 cal. yr. BP was obtained on the fourth outermost annual ring of a fossil tree with roots spreading into the underlying layer (Supplementary Table S4-2). Thus, the coastal zone could have been extensively flooded with seawater during the 12th century. The Pacific coast of NE Japan has been repeatedly hit by tsunamis with destructive waves [128]. Accordingly, the older tsunamis that invaded Lake Obuchi likely caused extensive sediment accumulation. However, deposits related to seawater flooding were not confirmed in the dune field except for the case of the 12th century tsunami.

The isotope anomaly was not identified in the Lake Obuchi core, owing to the lack of indicator benthic species (*A. beccarii*). However, the bivalvian-shell concentration is ascribed to rapid flows

from the shore. The high mud content, reaching up to 50%, was confirmed at the shelly horizon (Figure 2B), implying bottom erosion by surging currents. The nearly homogenous occurrence of heavy minerals throughout the sequence (Supplementary Figure S4) implies immediate particle settling from massive grain flows. The 2σ AMS age range of *A. beccarii* from the mud just below the lithological boundary (82–83 cm) is 470 ± 49 cal. yr. BP. (Supplementary Table S4-1, Figure S3A). The largest-scale tsunami caused by the subduction zone earthquake invaded the Pacific cost of NE Japan on December 2, 1611 [129]. It was not possible to estimate the depth range of the sediment removal. However, it is reasonable that the date of the bottom erosion lags fairly behind the AMS age. Thus, the inferred seawater inrush is ascribed to the 1611 tsunami. Although evidence of the 1611 tsunami has not been identified in the geologic and geomorphologic record, on the basis of ancient documents on tsunami hazards [129], the 1611 tsunami is inferred to have resulted in deep penetration, with massive seawater into the dune field through rivers. During that time, the aeolian sediment mobilization was declining under a wet and warm climate (Figure 8B,C), and the tsunami deposit might have been removed due to increased fluvial activities.

### 7.7. Effects of Undersea Dynamics on Dune Stability

Under modern tectonic conditions, the Shimokita Peninsula is subjected to E-W compressive stress. During the middle to late Holocene, however, the Pacific coast of the Shimokita Peninsula was scarcely subjected to tectonic elevation or subsidence (The Japan Atomic Power Company; https://www.nsr.go.jp/data/000153120.pdf), although geological evidence suggests the propagation of huge earthquake shocks in the study area (Figure 5). The paleotsunami records in eastern Hokkaido (Figure 1A) reveal that multi-segment earthquakes of about Mw 8.6 have occurred in the Kuril subduction zone at some time during the past [128]. A recurrence interval of approximately 400 years is interpreted for those earthquakes [130], with the most recent large event [131] corresponding to the 1611 Keicho earthquake [65]. The estimated date of the seismogenic event horizon preserved in Paleosol IV (early 17th century) is nearly equivalent to the historical age. The peninsula is close to eastern Hokkaido, and thus, it is likely that the major source of seismicity was located in the Kuril Trench (Figure 1A). Historical and observed tsunamis originating in the Kuril subduction zone were small in scale in the Shimokita Peninsula [128]. Thus, the tsunami that invaded the coast nearly 700 years ago probably did not originate in Kuril subduction zone.

The tsunami deposit of the 12th century is limited to the northern half of the dune field (Figure 1E), and a bed deposited by this tsunami event is not recorded in Lake Obuchi (Figure 2A). Sediment liquefaction/fluidization structures are missing in the Lake Obuchi core (Supplementary Figure S2). Thus, the localized flooding and deep penetration of seawater, at less than 5 km along the shoreline and more than 1.5 km inland from the beach, suggest the arrival of a tsunami derived by dislocation of the seabed.

Human activities have caused the abrupt growth of dunes and the resultant topographic modification in the Shimokita Peninsula since the 9th century (Figure 8B). Timber-felling to maintain fuel for iron manufacturing has repeatedly destroyed the forests in the dune field since the 9th century (Figure 8B), and the sand drifting over the exposed dunes has damaged the surviving trees [50]. This logging ended ~1870 AD [45], but the damage from blowing sand continued until ~1960 AD. Archaeological studies have revealed that industrial reactivation occurred after the recovery of forests [50]. Dunes III and IV were established during a time of extensive timber logging. It is, therefore, considered that human activities caused the abrupt growth of Dunes III and IV and the resultant topographic modification of the Shimokita Peninsula since the 9th century.

## 8. Conclusions

A well-developed field of Holocene aeolian dunes occupies the Pacific coast of the Shimokita Peninsula, northeast Japan. The concave-downward shape of foreset laminae suggests that the dunes are parabolic dunes. The dunes of the Shimokita Peninsula are divided into the following

four stratigraphic units: Dune I, ~6000–4000 cal. yr. BP; Dune II, 2000–1500 cal. yr. BP; Dune III, 1000–500 cal. yr. BP; and Dune IV, 150–60 cal. yr. BP. The chronological correlation of the reconstructed time-series of the geological processes denotes a one-to-one correspondence between the dune growth and the eustatic regression after the Holocene climatic optimum. Each dune unit grew in the course of regressive beach exposure. Each aeolian dune stratigraphic unit is overlain by a soil or paleosol. Since the Holocene climatic optimum (~6 kyr BP), tectonic movement has been negligible in this region, and episodes of aeolian dune growth have correlated with episodes of eustatic regression.

The aeolian dunes are resistant to erosion, and the development of terrestrial soils has been a major factor for dune stabilization and preservation. The progress of surface weathering depends on the meteorological and biochemical conditions in the region. A warm climate is the largest environmental factor for promoting soil formation, whereas precipitation acts directly on the dunes as a driving force for erosion. The Holocene climate change in the Shimokita Peninsula has been synchronous with climate changes in continental China. A reconstructed climate model indicates that the dune field of the Shimokita Peninsula was established under the control of the East Asian climate as a result of changes in hemispheric-scale atmospheric circulation. The paleosols that stabilized Dunes I and II developed under warm and wet conditions. Cool and dry climates are favorable for airborne grain motion on beaches, and thus, activated seasonal winds under these conditions enhanced the growth of Dune III. Intensified monsoons brought torrential rainfall in summer and high snowfall in winter. Heavy rains and massive snowmelt accelerated surface erosion and fluvial sediment discharge. The climatic warming and resultant additional precipitation under monsoonal intensification promoted the increased influx of fluvial sediments, and the enriched sand budget for the beach contributed as the secondary cause and prerequisite for establishing Dunes I and II. Forest-felling to maintain fuel for iron manufacturing caused frequent damage to the vegetation cover in the Shimokita Peninsula. Deforestation also caused aeolian grain drift over the exposed dunes, resulting in additional casualties to the surviving trees. Therefore, the industrial activity acted as a secondary factor influencing the growth of Dunes III and IV. Full-scale afforestation began in the Shimokita Peninsula in 1930 AD. Since then, sand drift has been controlled, and Dune IV has retained its original form.

Northeast Japan is susceptible to subduction-zone earthquakes, and seismic shocks trigger the gravitational collapse of aeolian dunes on various scales. Dune-slope dislocation has dammed watercourses with slumped debris, resulting in the submergence of fluvial valleys. Fluvial sediments, derived from alluvial fans to the west of the coastal dune field, have covered the bottom of submerged valleys and the sediment-filled areas have become forested with conifers. The earthquake shocks have acted as the driving force leveling the topographic relief through aggradation.

**Supplementary Materials:** The following are available online at http://www.mdpi.com/2076-3263/10/10/410/s1, Figure S1: Optical photomicrographs of representative fossil diatoms, Figure S2: Outcrop photograph showing the deposition of organic soil (Paleosol III), medium sand with abundant mafic grains, and trough cross-bedded feldspathic sand (Dune III) in ascending order; Figure S3: Vertical section showing the stratigraphic succession of Dune III, Paleosol IV, and Dune IV in ascending order; Figure S4: Stratigraphic changes in grain mineralogy of the sand layer occupying the depth interval between 81.5 and 40.5 cm (Unite 4) of the Obuchi core; Figure S5: SEM photographs of foraminifera originating in the core of Lake Obuchi; Figure S6: Optical photomicrographs of representative fossil pollen from the core drilled in Tashiro Wetland; Table S1: Mineralogical results of the samples obtained from the stratigraphic interval between 82 cm and 40 cm of the Obuchi core; Table S2: Occurrence list of foraminifera fossils from the stratigraphic interval between 90 cm and 0 cm of the Obuchi core; Table S3: Stable isotopic results of perfect fossils (Table S3-1) and living (Table S3-2) tests of *A. beccarii* extracted from the sliced sediments of the Obuchi core; Table S4-1: AMS age data of the perfect fossil tests of *A. beccarii* from the Obuchi core; Table S4-2: AMS age data of the second- and fourth-outermost annual rings of a standing fossil conifer (*Thujopsis dolabrata*) from Tatenuma and Zaimokuzawa (Figure 1D); Table S5: Palynological results of arboreal (43 genera) and herbaceous (34 genera) fossil pollens from the Tashiro core; Table S6: Calculated results of climate parameters (annual mean temperature and annual precipitation) on the basis of the BMA method adopted for the pollen data in Supplementary Table S5.

**Author Contributions:** Conceptualization, K.M.; methodology, K.M.; software, K.M.; validation, K.M. and N.N.; formal analysis, K.M.; investigation, K.M.; resources, K.M.; data curation, K.M.; writing—original draft preparation, K.M. and N.N.; writing—review and editing, N.N.; visualization, K.M.; supervision, K.M.; project administration, K.M.; funding acquisition, K.M. and N.N. All authors have read and agreed to the published version of the manuscript.

**Funding:** This research received no external funding.

**Acknowledgments:** The authors would like to thank T. Yamanoi and K. Hayashibara for assisting in the pollen identification, N. Nemoto at Hirosaki University for obtaining the foraminiferal data, and A. Yoshida and T. Yamada at the Tashiro Wetland for the core recovery. The authors are very grateful to the invaluable comments and suggestions provided by three anonymous reviewers. The International Research Institute of Disaster Science, Tohoku University, financially supported the coring in Lake Obuchi. We would like to thank www.editage.com for English language editing.

**Conflicts of Interest:** The authors declare no conflict of interest.

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
