# Peer review of "Eustatic, Climatic and Tectonic Controls on the Evolution of a Middle to Late Holocene Coastal Dune System in Shimokita, Northeast Japan"

_geosciences, doi:10.3390/geosciences10100410_

Round 1

Reviewer 1 Report

This manuscript is very, very difficult to review because (1) the writing style is not succinct, and (2) the organizational structure of the manuscript is unnecessarily complicated.  New information is presented to the reader constantly at random places throughout the manuscript.  The authors must do better at separating (1) descriptions; (2) interpretations; and (3) discussion points.  Their new data consists primarily of the following three items: (1) descriptions of exposures along rivers in a coastal eolian dunes field; (2) a core taken from a lake near the coastal eolian dune fields; and (3) a core taken in a highland area ~60 km southwest of the coastal eolian dune field.  The authors really need to provide succinct descriptions of each site, then provide succinct interpretations of each, and then discuss greater implications of their descriptions and interpretations.  In the attached pdf, I have made numerous suggestions for restructuring the manuscript and making the text more succinct.  These structural problems with the text should have been addressed before the manuscript was submitted for review.  This is the second time that I have reviewed this manuscript, and both times it has been absolutely mind-numbing work for me to look at each sentence and make suggestions about where in the text it belongs.  This burden of work is more than what should be asked of a typical manuscript-reviewer.  Nevertheless, I really would like to see this material published.  The data and conclusions are very interesting -- but if the organizational structure of the manuscript cannot be fixed, then readers will not pay attention to the manuscript because they will not want to work so hard to separate descriptions, interpretations, and discussion points.  

Author Response

We appreciated your very thoughtful comments. We reorganized our manuscript as a whole to answer your whole comments by taking back to the raw descriptions of the field surveys of geology and core descriptions as well. We believe that these revisions are met with your comments. 

Very sorry for the late reply. We had needed time to answer your comments.

Reviewer 1

Replies to the comments from reviewer 1 by point-by-point.

1, According to his/her comment, we change the title to “Eustatic, climatic, and tectonic controls on the evolution of a middle to late Holocene coastal dune system in Shimokita, Northeast Japan.”

2, According to the reviewer’s comment, we change the Abstract in lines of 10-21.

3, According to the reviewer’s comment, we added “Japan” in Keywords.

4, According to the reviewer’s comment, we added references in lines 27, 30, 35, 41, 45.

5, According to the reviewer’s comment, we modified the Introduction as follows:

 5a) making a separate paragraph about Holocene in lines 42-51.

 5b) making a new paragraph to summarize this study in lines 52-60.

 5c) modifying a figure caption

6, According to the reviewer’s comment, we modified the Study area as follows:

 6a) modifying the first paragraph in lines 63-71.

 6b) making a separate paragraph about the modern climate/meteorology of the dune field area of the Shimokita Peninsula in lines 72-83.

 6c) making a separate paragraph about the Lake Obuchi in lines 84-87.

 6d) making a separate paragraph about the Tashiro Wetland in lines 88-103.

 6e) making a separate paragraph about the modern climate in Shimokita Peninsula in lines 104-109.

 6f) making a copied & pasted paragraph from Discussion in lines 110-119.

7, According to the reviewer’s comment, we modified the Methods by moving lots of sentences from section 5 inline 122-199.

8, According to the reviewer’s comment, we reconstructed the Results to Depositional Environments of the Shimokita Peninsula as follows:

 8a) making sub-section of 4.1 as Beach environment in lines 205-220.

 8b) making sub-section of 4.2 as Coastal eolian dunes in lines 222-291.

 8c) modifying Figure 2

 8d) making a new detailed description of field outcrops (e.g., four units) in this area in lines 236-264.

 8e) making sub-section of 4.3 as Lacustrine systems in detail in lines 294-328.

 8f) making sub-section of 4.4 as Fluvial systems

 8g) changing the word “sand drape” into “mud drape” that is covering the sedimental surface on the dune by mud deposited during slack water inline 346.

 8h) describing the numbers for thicknesses, elevations and distances.

 8i) making sub-section of 4.5 as Alluvial systems in detail in lines 376-381.

9, According to the reviewer’s comment, we created a new section of the Core description in lines 383-504. We presented the core descriptions by being back to our fieldnote raw descriptions. Also, we moved some of the sentences to various sections of Methods, Interpretations and Discussion. Also, we modified Figure 3 by putting explanations of units in the columns.

10, According to the reviewer’s comment, we created a sub-section of 5.1 as Core took from Late Obuchi in lines 384-484 in detail with detailed quantitative descriptions.

11, According to the reviewer’s comment, we created a sub-section of 5.2 as Core took from the Tashiro Wetland in lines 486-504 in detail with detailed quantitative descriptions.

12, According to the reviewer’s comment, we created a new section of 6 as Interpretation of the Outcrops/Exposures and the Cores in lines 506-682.

  12a) we wrote some paragraphs about sedimentary structures in Dine III and Paleosol IV in lines 5-7-584.

  12b) we wrote some paragraphs about Obuchi core in lines 586-652.

  12c) we wrote some paragraphs about The Tashiro Wetland in lines 654-673.

13, According to the reviewer’s comment, we reorganized the section of Discussion as follows:

  13a) making a sub-section of 7.1 Climatic influence on dune growth in lines of 677-698.

  13b) making a sub-section of 7.2 Holocene paleoclimate reconstruction in lines of 701-725.

  13c) making a sub-section of 7.3 Atmospheric background on local vegetation in lines of 727-762.

  13d) making a sub-section of 7.4 Meteorological impacts on dunes in lines of 764-774.

  13e) making a sub-section of 7.5 Global and local control on dune growth in lines of 776-816.

  13f) making a sub-section of 7.6 Effects of marine events on eolian dune landforms and coastal environments in lines of 818-849.

  13g) making a sub-section of 7.7 Effects of undersea dynamics on dune stability in lines of 851-878.

Reviewer 2 Report

Authors have responded accordingly to my initial review concerns regarding grammar. I recommend acceptance in its present form.

Author Response

Thank you for your review.

Reviewer 3 Report

The authors did a very good job in revising the paper by taking some of my comments into account.

Although it is not a problem if one or the other comment remain unresponded, it is very important to appropriately acknowledging previous work of researchers who did important contribution to the field. In a recent paper, Gabarrou et al. discussed the role of sand availability for the formation of coastal sand dunes in detail, by coupling numerical simulations using realistic winds with satellite and field observations:

Gabarrou et al. (2018), Modelling the Retreat of a Coastal Dune under Changing Winds", Journal of Coastal Research 85 (sp1), 166-170.

Therefore, it is fair and correct to include this reference, although I understand that the authors may not be comfortable enough to draw comparisons between their work with the referred paper.

However, the authors should include the aforementioned citation for instance in the sentence at line 36 on page 1: "Thus, the abundant sand supply from the surf zone increases the potential of dune development [...]".

If this citation is included, then I have no objection to publication and will recommend acceptance - I don't need to see the paper again, if the authors include the citation the paper can be published as is.

Author Response

Thank you very much for your review.

According to your comment, we added references in lines 34.

I hope your suggestion would be involved in the text.

Nori

Round 2

Reviewer 1 Report

I would like to see this manuscript published, but this has been an extraordinarily difficult manuscript to review.  The major problems are related to the structure of the manuscript, the order in which information is presented to the reader, and an inadequate separation of descriptions from interpretations.  As a typical example of such problems: In the Discussion section (Section 7) of the manuscript, the authors discuss the estimated date of a "seismogenic event horizon."  This is the first place in the manuscript that the reader encounters this phrase "seismogenic event horizon," and there is no context provided for the reader to understand what this feature is.  To fix this problem, something needs to be described in Section 4 (Descriptions), and then this something needs to be interpreted as a "seismogenic event horizon" in Section 6 (Interpretations).  Only after this context is provided, the reader might understand what is being discussed in the "Discussion" section.  In addition the lack of topic sentences sometimes makes it difficult for the reader to grasp the subject or theme of given paragraph.

Although the revised manuscript is much better than the previous version that I reviewed, the manuscript still has several inconsistencies with the Descriptions and Interpretations.  The manuscript is also extremely verbose, and many of the sentences are constructed in an awkward manner.  I have made numerous suggestions for improvement (see attached pdf).  If my suggestions are addressed, then I think that the manuscript could be ready for publication.  Again, I would like to see this manuscript published, but it still needs some improvement.

Author Response

We appreciated your very thoughtful comments on Round 2. Based on your commented manuscript, we modified the whole MS and Figures. We believe that these revisions are met with your comments. 

Reviewer 1 on Round 2

Replies to the comments from reviewer 1 by point-by-point. The following line numbers are in tracking mode.

1, According to the reviewer’s comment, we changed eolian into “aeolian” in lines 11. The whole eolian was changed.

2, According to the reviewer’s comment, we changed the abstract in lines of 12-16, 20-24.

3, According to the reviewer’s comment, we added “Holocene” in Keywords.

4, According to the reviewer’s comment, we changed the texts in lines 58, 59, 60, 66, 67, 68, 69.

5, According to the reviewer’s comment, we changed sandhills into dunes inline 73.

6, According to the reviewer’s comment, we changed texts in lines 74, 77, 79, 82, 83, 84, 88, 89, 90, 91, 92, 94, 95, 96, 97, 106, 107, 110, 112, 113, 115, 120, 122, 125, 126,128, 129.

7, According to the reviewer’s comment, we modified the last paragraph of 2. Study area.

8, According to the reviewer’s comment, we modified texts in lines 138, 148, 154, 156, 158, 180, 184, 185, 186, 188, 195, 204, 208, 210-217.

9, Based on the reviewer’s comment, we changed the text in lines 222, 224, 225, 228-237, 241-244, 246-250, 252, 254, 255-264, 266, 268, 270-272, 274-325.

10, According to the reviewer’s comment, we changed the texts in lines 324, 325, 331, 332, 333, 337, 338-345, .

11, According to the reviewer’s comment, we changed the texts in lines 349, 350-358, 360,361, 363, 364, 365-388.

12, According to the reviewer’s comment, we modified sub-section 4.4 in lines 391-456.

13, According to the reviewer’s comment, we modified sub-section 4.5 in lines 458-464.

14, According to the reviewer’s comment, we modified section 5 in lines 466-644.

15, According to the reviewer’s comment, we modified section 6 in lines 648-918.

16, According to the reviewer’s comment, we modified section 7 in lines 924, 931, 935, 938, 939, 942-946, 950, 954, 956, 957, 958, 959, 960, 961, 962, 963, 964, 965, 978, 979, 980, 981, 982, 983, 984, 985, 995, 997, 1001, 1002, 1003, 1004, 1007, 1008, 1009, 1011, 1013, 1021, 1022, 1031, 1034, 1036, 1038, 1039, 1040, 1043, 1047, 1050-1056,1057, 1073, 1078, 1107, 1112,1113, 1117, 1122,1123,1124, 1125, 1127, 1128, 1130, 1131, 1132,1133, 1136, 1137, 1139, 1140, 1141.

17, According to the reviewer’s comment, we modified section 8 in lines 1144-1184.

18, According to the reviewer’s comment, we added the location of the Towada volcano in Fig. 1.

19, According to the reviewer’s comment, we changed Fig. 3 and Fig. 4 to explain stratigraphic units.

//

This manuscript is a resubmission of an earlier submission. The following is a list of the peer review reports and author responses from that submission.

Round 1

Reviewer 1 Report

This manuscript is not ready for publication.  There are some major structural problems with the manuscript -- Descriptions are not clearly separated from interpretations, important descriptive data are presented to the reader in unexpected sections of the manuscript, and in some places it is not clearly articulated what work was done by the authors and what work was done by others.  In the attached MSWord file, I have made a lot of suggestions for improvement with regards to these issues.  In addition, the manuscript is exceptionally verbose, especially the "Discussion" section, and could be shortened and focused greatly.  Again, I have made many suggestions.  I have also suggested some additional aeolian-related publications that the authors might want to consult, and I have made some suggestions for general improvement of the English language.  My overall impression, however, is that this manuscript presents some good work, and I would like to see this material published.  I suggest that the authors work with my suggestions, and then re-submit the manuscript.  I am willing to review the manuscript again, if the journal editors decide to follow my recommendation.

Author Response

We would like to thank reviewer for your valuable comments to improve our manuscript. We have revised our manuscript according to your comments. Please check them below.

To Reviewer 1

(1) Description of the modern coastal eolian dune field

  → 4. Results; 4.2. Dune morphology and structure (p. 4 line 38–42)
(2) Description of the modern fluvial systems of the study area
  → 4. Results; 4.3. Modern fluvial systems (p. 4–p.6)

(3) Description of the modern lacustrine and marsh systems of the study area
  → 4. Results; 4.1. Modern beach environment (line 174–179)

(4) Description of the modern beach environment

  → 4. Results; 4.1. Modern beach environment (p. 4 line 26–31)

(5) Description of the core taken from Lake Obuchi

  → 4. Results; 4.4. Description of the coastal lake core (p. 6 line 23–-29)

(6) Description of the core taken from the Tashiro Wetland

  → 4. Results; 4.5. Description of the crater lake core (p. 7 line 33–42)

What is the size of the dune field?

  → 4. Results; 4.2. Dune morphology and structure (p. 4 line 38–39)

What is the relief on the dunes?

  → 4. Results; 4.2. Dune morphology and structure (p. 40 line 40 – 42)

What are the shapes of the dunes?

  → 4. Results; 4.2. Dune morphology and structure (p. 5 line 3–13)

What are the inter-dune areas like, and are they filled with sediment? What lies beneath the dunes?

  → 4. Results; 4.2. Dune morphology and structure (p. 2 line 39–42, p. 4 line 39–40, p. 5 line 20–23, Cross sectional view in Figure 2)

What sedimentary structures and what stratigraphy are visible within the dunes?

  → 4. Results; 4.2. Dune morphology and structure (p. 5 line 6–14)

In what compass directions does the cross-bedding dip?

  → 4. Results; 4.2. Dune morphology and structure (p. 5 line 11–12)
What is the composition of the dunes? – grain size, sorting, lithology, quantitative color data, etc.

  → 4. Results; 4.2. Dune morphology and structure (p. 5 line 4–5, 7–9)

Reviewer 2 Report

I found this manuscript to be very high quality with excellent graphics and imagery. The study is multi-faceted but well integrated, with good discussions clearly combining the significance of the different data sources. The conclusions are well supported by the evidence. 

Author Response

We would like to thank reviewer for your valuable comments to improve our manuscript. We have revised our manuscript according to your comments. Please check them below.

To Reviewer 2

(1) →  1D (wind rose on Google image)

(2) →  2 (geological section)

(3) →  Methods; 3.1. Aeolian and paleosol deposits (p. 2 line 42, p. 5 line 1–2)

(4) →  Study area; (p. 2 line 24–35)

(5) →  Results; 4.2. Dune morphology and structure (p. 4 line 40–42, p. 5 line 1–4.)

Reviewer 3 Report

This is an impressive, very detailed study about the geologic history and dynamics of dunes in Shimokita Peninsula, Japan.

These dunes are not discussed often in the literature. The manuscript is illustrated with fantastic figures, and is very well written. The conclusions have impact to understand dune dynamics, the feedbacks between climate and sediment transport, and the climate history of Japan. The paper is so great and represents such a great contribution to Aeolian Geomorphology that I can recommend very strongly to highlight it in the front page of the Geosciences (MDPI). “Best papers of the month”, or something similar.

Nevertheless, before publication, there are important details which the authors should add/correct.

The title is misleading because the authors also consider seismic influences on the dunes. Does seismic activity belong to “climate influence”? What about including “geologic factor” or something similar in the title? Moreover, the authors also consider anthropogenic influences. These don’t belong to climate or oceanic impacts. So please make the title more adequate in view of what you are presenting in the paper.

In Section2: Please specifiy what is the classification of dunes in Shimokita peninsula (barchans? parabolic dunes? Transverse? Linear? dome dunes? nebkhas?). Or is it not possible to classify them?

Also in Section 2, the authors should show a satellite or aerial photo of the dunes, and/or pictures of the dunes taken (by the authors) in the field. This would help the reader to follow the paper.

In the satellite/aerial photo the authors should identify dunes I, II, III and IV as they did in Fig. 1.

Please add to Fig. 2 or in a separate Figure the sand rose from the climate station that best reflects the climate in the dune field. You can refer to Ref. 10 (paper by Haim Tsoar) to compute the sand rose. Haim Tsoar has the sand rose of most dune fields in the world, and I can recommend you to contact him if you need assistance with the sand rose calculation.

In Section 5.1, first paragraph: How do you know that these are mega-ripples and not small transverse dunes? We know that the minimum size of transverse dunes is about 50cm - 1 m.  Therefore, couldn't the bedforms be small transverse dunes? Please discuss and should show photos of these bedforms that you have classified as mega-ripples.

One hint: If these bedforms have a bimodal sand distribution, with very large grains on their crests with fines on the throughs and in the inner layers,  then they are certainly mega-ripples, otherwise there is nothing obvious to me for why they should be classified as something different from transverse dunes.

Could you please show a graph of the yearly behavior of rainfall and wind power (and/or direction)? We know that dune dynamics respond largely to seasonal variations in rainfall and wind. Therefore, this information of the seasonal variability should be included. Moreover, in the discussion sections, I couldn’t perceive what is the role of seasonal rainfall and wind variations for the Shimokita dunes and vegetation cover (beyond the evolution over the millennia). Could you please attempt to discuss a bit more explicitly about this role?

Could the authors cite, as one further example of study on coastal dune fields, the recent work by Gabarrou et al., “Modelling the Retreat of a Coastal Dune under Changing Winds”, Journal of Coastal Research 85 (sp1), 166-170 (2018)? These authors combined remote sensing and field data with modelling to investigate the retreat of Dune du Pilat, which is the largest European dune (100 m high) and develops in the presence of vegetation. I would find great if the authors could also briefly mention what are the main differences between the environmental conditions in Shimokita dune field and at dune du Pilat, which make these dunes so different. More precisely, an interesting question for the non-specialist is the following: Why haven’t Shimokita dunes become a “dune du Pilat”?

Author Response

We would like to thank reviewer for your valuable comments to improve our manuscript. We have revised our manuscript according to your comments. Please check them below.

To reviewer 3

(1) →  Methods; 3.1. Eolian and paleosol deposits (p. 2 line 42, p. 5 line 1–2)

(2) → 4.4. Meteorological impact on dune growth (p. 14 line 24–33)